# Understanding the effects of real-time head position feedback on postural sway in terms of changes in underlying deterministic and stochastic dynamical processes

Steven J. Harrison[1,2]*, Benjamin De Bari[3], Ryan Poutre[2], Jeffrey M. Kinsella-Shaw[1,2]

1 Center for the Ecological Study of Perception and Action, University of Connecticut, Storrs, Connecticut, United States of America, 2 Department of Kinesiology, University of Connecticut, Storrs, Connecticut, United States of America, 3 College of Humanities and Sciences, Thomas Jefferson University, Philadelphia, Pennsylvania, United States of America

* steven.john.harrison@gmail.com

## Abstract

Our perceptual systems detect information about how our body is moving relative to the surrounding environment. Such information is essential to our ability to maintain upright standing balance. Real-time postural feedback systems are technologies that appear to enhance our ability to detect information about the movements of our body, and as a consequence improve our capacity to control the posture of our body. Here we examine a system in which real-time measurements of head position were "felt" as changes in the intensity of vibration of tactors in a headband. Using this system, participants were able to stabilize their body posture, showing a reduction in the magnitude of head sway fluctuations of more than 40% under single and dual tasking conditions. To examine this effect, we used a dynamical model that assumes that observed magnitudes of postural fluctuations reflect combined effects of underlying deterministic and stochastic dynamical processes. We derived estimates of model parameters, $\lambda$ and $Q$. $\lambda$ measures attractor strength, it quantifies the influence of the deterministic process that cause postural fluctuations to tend to drift towards a desired state. $Q$ measures noise, it quantifies the destabilizing effects of internal perturbations and background/physiological variability in the postural control system. We find that use of our head position feedback system led to marked increases in attractor strength. This increase in attractor strength was accompanied by a marked increase in noise. When we linearly increased the resolution of feedback—by narrowing the width of the tactor activation function—we observed an approximately linear increase in attractor strength together with a non-linear increase in noise. These results suggest that use of a real-time postural feedback systems can increase the stability of postural control, but that this benefit is limited by significant accompanying exacerbations of noise being introduced into the postural control system.

**Data availability statement:** The kinematic data that formed the basis of the analyses presented in this paper are available through OSF (DOI . https://doi.org/10.17605/OSF.IO/KR2YM).

**Funding:** The author(s) received no specific funding for this work.

**Competing interests:** NO authors have competing interests.

## Introduction

Our capacity to effectively control the posture of our body is essential to our ability to perform most everyday actions [1,2]. Both the *form* of postural control (i.e., the way in which the degrees of freedom of the body are coordinated) and the *function* of postural control (i.e., how effectively the posture of the body is controlled in the service of performing an intended action) are shaped by the interactive effects of various *sources of constraint* [3–6]. Sources of constraint include task constraints (e.g., the task demand of maintaining a stable head position while standing and reading some text in the environment) [7,8], motor constraints (i.e., the factors that determine the forces that can be directed to stabilize the posture of the body) [9–11], and informational constraints (i.e., the information about the body movement detected by the visual, haptic, and vestibular perceptual systems that is used to perceptually guide the control of posture) [12–15].

The effects of sources of constraint upon postural control are evident in the systematic effects they have upon postural fluctuations [7,13,16]. Commonly studied postural fluctuations include body sway (e.g., movements of the head or the center of mass of the body in space) and center of pressure excursions (i.e., changes in the locus of the reaction force vector on the support surface) [2,17]. Changes in informational constraints are associated with pronounced changes in the magnitudes of postural fluctuations; with fluctuations tending to be far greater when the information detection capacities of an individual are impaired/absent or when the environmental conditions supporting the effective detection of information are impoverished [13,18,19]. For example, when participants have been asked to stand in a tandem stance (i.e., one foot in front of the other) with their eyes closed, the added constraint of allowing one hand to lightly touch an adjacent surface (i.e., providing a rich additional source of haptic information) has been shown to reduce the magnitudes of postural fluctuations in the medial-lateral direction by 66% [16].

### Using real-time postural feedback as an informational constraint

The potential of our perceptual systems to supply informational constraints that dramatically increase the stability of postural control has led engineers and researchers to explore the possibility of creating new sources of informational constraint using technology [20–22]. A common basic design for such technology uses sensors to measure some aspect of the body posture (e.g., trunk lean) and a "display" of some kind to relay the information detected by that sensor in real-time to the user [21]. It has been repeatedly shown that the use of real-time posture feedback by heathy adults and individuals with impaired balance can significantly reduce the magnitude of postural fluctuations [21] and can improve balance recovery responses following unexpected perturbations [23].

The basic effect of reducing the magnitude of postural fluctuations through the aid of a real-time posture feedback system is observed regardless of whether the type of display used in the system is visual [24,25], auditory [26,27], electrotactile [28,29], vibrotactile [20,23], or multimodal [30]. Some researchers have expressed a preference for using vibrotactile displays for real-time posture feedback systems based

upon the logic that vibrotactile feedback can supply a source of balance-specific information without interfering with the important functions sight or hearing [31]. Previously studied real-time posture feedback systems have displayed varied forms of real-time information, including information about body tilt angle [24], body sway velocity [32], body sway acceleration [33] and center of pressure location [25]. Larger reductions in the magnitudes of postural fluctuations have been observed when information about body position (rather than body velocity) is provided [32].

### Carry-over effects from experience using real-time postural feedback systems

Some researchers have observed that experience using a real-time posture feedback system can have lasting effects, such that when the device is switched off, alterations to the form and function of postural control can be observed. Ballardini et al. [33] examined possible carry-over effects resulting from the use of a feedback system that provided real-time information about body sway acceleration to young healthy adults. While switched on, this device reduced the magnitude of body sway accelerations and altered the frequency profile of those accelerations. When this device was switched off, the observed changes in the frequency profile were retained in a carry-over trial, suggesting that participants had altered how they were controlling their body posture, at least temporarily. Relatedly, experience using real-time posture feedback systems by individuals with impaired balance, have shown both short-term and long-term carry-over effects [21]. For example, Tyler et al. [29] observed short term carry-over effects resulting from the use of a feedback system that provided real-time information about head position to individuals affected by bilateral vestibular dysfunction. While switched on, this device drastically reduced the magnitude of head sway fluctuations. These reductions in head sway fluctuations were retained for as much as 100 s after the device was switched off.

### Vibrotactile displays

In vibrotactile displays, information about the posture of the body is relayed through vibrating elements, or "tactors" (Fig 1A) that can be embedded in a garment (e.g., a belt or headband). In vibrotactile displays, information about the direction of body sway is typically conveyed through the location on the body at which tactor vibration is felt. If a large array of tactors wrapped around the torso or head, it is possible to convey information about sway in multiple directions [23,32,34,35]. Alternatively, using just two tactors, it is possible to convey information about body movement in a particular sway axis (e.g., vibration of tactors positioned on the belly and back generating information about anterior and posterior

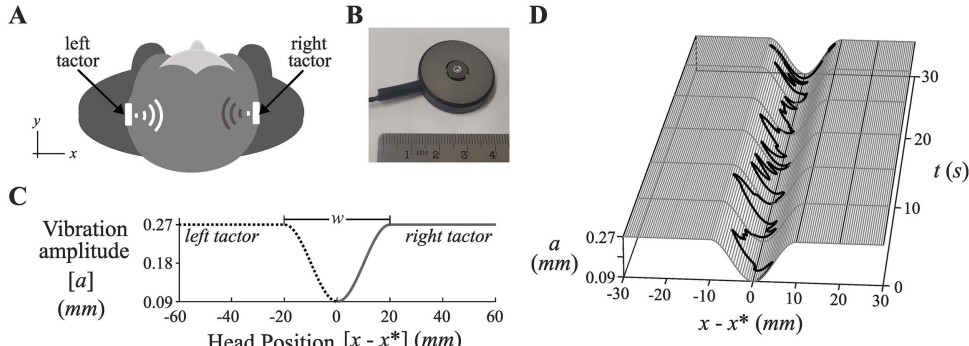

**Fig 1. Real-time head position feedback.** A) Tactors positioned on the left and right side of the head produced patterns of vibration that varied with medial-lateral head sway. B) One of the C-2 tactors used in the experiment. C) The tactor activation function showing the relationship between head position ($x$) measured relative to a predetermined set point ($x^*$) and the amplitude of vibration of the left tactor (dotted line) and right tactor (solid line). The parameter $w$ is the trough width of the function. A parameter setting of $w = 40$ mm is depicted. D) An example of a trial in which head position feedback was provided with $w = 20$ mm. The plot shows head position as a function of time ($t$) and tactor vibration amplitude ($a$).

sway respectively) [24,33]. Vibrotactile displays can convey information about the magnitude of sway in a given direction through changes in the intensity (i.e., frequency and/or amplitude) of tactor vibration [24,33].

## The dynamics of postural fluctuations

While it is commonplace to examine the magnitudes of postural fluctuations in investigations of real-time posture feedback systems, researchers have not examined how use of real-time posture feedback affects the *dynamics* of postural fluctuations. The dynamics of postural fluctuations refers to the processes governing the spatiotemporal trajectory of fluctuations [13,14,36–40]. The dynamics of postural fluctuations have been captured with models that assume observed fluctuations represent the combined effects of underlying deterministic and stochastic processes [41–45]. Of these models, the Ornstein-Uhlenbeck model is one of the simplest and has been shown to account for high percentage of the variance of postural fluctuations measured during standing balance [41,45]. The Ornstein-Uhlenbeck model is a model of a stochastic process constrained to drift towards an equilibrium position [43,44]. It has just two parameters. The first model parameter, $\lambda$, describes the strength of a drift process. For this parameter, negative values indicate the existence of a state (e.g., a location in space) that postural fluctuations will have a tendency to drift towards. In the parlance of dynamical systems theory, negative $\lambda$ values represent the existence of a fixed point attractor, and more negative $\lambda$ values represent greater attractor strength. The second model parameter, $Q$, describes the strength of a diffusion (random walk) process. Larger $Q$ values indicate greater fluctuating forces (e.g., intrinsic noise) in the system and a faster diffusion process [41]. In the parlance of dynamical systems theory, $Q$ values indicate the noise in the system, where noise refers to "dynamical noise" rather than measurement noise and relates to the physiological variability that is believed to be an intrinsic property of the system dynamics [46]. The identification of $\lambda$ as a measure of attractor strength and $Q$ as a measure of noise as key system parameters in the Ornstein-Uhlenbeck model parallels the choice of some researchers to focus upon measures of *maximum line length* and *percent recurrence* when performing recurrence quantification analysis (RQA) based measurements of postural fluctuations [13,14,38]. In RQA *maximum line length* has been interpreted as a measure of attractor strength, and the inverse of *percent recurrence* has been taken as a measure of noise [13,47].

As with simple measures of the magnitude of postural fluctuations, measures of the dynamics of postural fluctuations are known to be systematically affected by changes in task, motor, and informational constraints [13,39,45]. Bonnet et al. [41] observed that increasing the richness of the visual environment (i.e., standing in an environment that enhances the capacity to visually detect body sway) was associated with a decrease in estimates of noise in young heathy adults, and that individuals with greater visual detection capabilities (i.e., visual contrast sensitivity) had greater values for their estimates of attractor strength. They also observed that estimates of noise were greater for older participants, and that attractor strength estimates depended upon changes in task (i.e., standing while performing a text scanning task vs. just looking straight ahead). Relatedly, Kinsella-Shaw et al. [13] found that RQA-based estimates of noise increased when the level of room illumination was decreased, RQA-based estimates of attractor strength increased with both the level of room illumination, and the richness of the visual surroundings. In sum, these findings suggest that the availability of rich sources of visual information, and the capacity of an individual to detect that information can increase stability of postural control (i.e., measures of attractor strength) and decrease the inherent variability of postural control (i.e., measures of noise). Additionally, the influence of the availability of visual information for postural control was also observed to interact with intrinsic age-related differences in the relationship between attractor strength and noise.

Quantifications of the dynamics of postural fluctuations can reveal the effects of sources of constraint more subtle than age. For example, Pellecchia and Shockley [48] observed that estimates of attractor strength derived from measurements of center of pressure fluctuations during stable standing decreased when participants were given an additional "dual" task of counting backwards by 3's from a random starting number.

## Current study

The current study was motivated by the goal of advancing our understanding of real-time postural feedback systems by examining how the provision of such feedback affects the dynamics of standing balance. In other words, we aim here to better understand the informational constraints that are created when a person uses a real-time postural feedback system. How do these informational constraints affect the stability and intrinsic noise of a postural control dynamic? How do the informational constraints obtained from a real-time postural feedback systems differ (or not) from those obtained from "natural" informational constraints such as vision?

In the current study, we examined how real-time feedback of medial-lateral head position (Fig 1) would affect both the magnitude and dynamics of body sway fluctuations for young healthy adults instructed to "stand as still as possible". To facilitate our interpretation of the effect of real-time postural feedback we included two standard baseline conditions: a standing with eyes open control condition (EO), and a standing with eyes closed control condition (EC). These conditions were compared to a condition in which participants stood with their eyes closed while having available real-time feedback of medial-lateral head position (EC + F$_{ML}$). Across all three of these conditions, participants tried to maintain a posture with minimal head movement. Thus, the overarching intention across these conditions was to maintain a maximally stable head orientation to the environment.

During all standing balance trials stood with a narrow base of support, feet together symmetrically relative to the midline of the force platform. This postural (motor) constraint was introduced to minimize the range over which sway could occur before destabilizing a participant's posture, moving them away from the intended head orientation. Standing in a symmetrical, "feet together" stance is a common component of clinical balance testing. A narrowed stance, reducing the base of support's dimensions in the frontal plane has been shown to elevate control demands of stabilizing posture under a variety of conditions, including in the absence of visual support. Given that the participants were all healthy, young adults with no histories of balance impairments, this level of challenge was imposed to increase the likelihood that providing additional information for control, i.e., real-time vibrotactile feedback, would yield measurable differences in sway dynamics and magnitudes.

We predicted that the availability of visual information in the EO condition—compared to the absence of such information in the EC condition—would lead to 1) decrease the magnitude of body sway fluctuations [13,37,49], and 2) affect the dynamics of body sway fluctuations by increasing attractor strength ($\lambda$) and/or decreasing noise (Q) [13,49].

We theorized that the availability of real-time head position feedback in the EC + F$_{ML}$ condition would affect the magnitude and dynamics of postural fluctuations in the same way that the availability of rich sources of visual information has previously been shown to [13,41]. Specifically, we hypothesized that the availability of head position feedback would 1) decrease the magnitude of body sway fluctuations, and 2) affect the dynamics of body sway fluctuations by increasing attractor strength ($\lambda$) and/or decreasing noise (Q). We hypothesized that these predicted effects would occur only in the direction of sway for which the feedback was given [33,50]. Specifically, we predicted that our real-time feedback of medial-lateral head position would affect medial-lateral sway and would not affect anterior-posterior sway.

We provided feedback of medial-lateral head position using tactors positioned above the ears on the left and right side of the head (Fig 1A). Our choice to examine this positioning, rather than an anterior-posterior tactor placement, was motivated in-part by a wish to avoid the complications of introducing potential effects of vibration sensitivity asymmetries into our experiment. Vibration sensitivity asymmetries would likely have existed had we positioned the tactors on the anterior and posterior surfaces of the body [51,52]. The potential to modify postural sway in the medial-lateral direction is of particular interest to us for the following reasons: Changes in medial-lateral sway control has been reported as more predictive of an elevated risk of falls in older adults and neurologically impaired populations (e.g., patients with multiple sclerosis) compared to changes in anterior-posterior sway control [53–57]. The effective regulation of medial-lateral body sway is known to be important for supporting basic human functions such as transitioning between standing and walking (i.e., step initiation in bipedal gait requires controlled lateral weight shifting) [58,59] and precise aiming [60]. Lastly, even healthy

younger adults, studied here, whose risk of injurious falls is considered minimal have been observed to show increased medial-lateral sway variability when visual support is significantly diminished or excluded [61,62].

We hypothesized that parameterizations of the provided feedback—achieved by varying the trough width (*w*) of the tactor activation function (Fig 1C)—would affect both the magnitude and dynamics of body sway fluctuations. We predicted that reducing trough width would both decrease the magnitude of body sway fluctuations and increase the attractor strength in the dynamics of postural fluctuations.

Haggerty et al. [31] studied the standing balance of community dwelling older adults using a real-time posture feedback system. They examined how the aid to balance provided by the feedback system was affected by having participants concurrently perform a secondary task. Although the stabilizing effect of the feedback was not diminished by the secondary task, performance at the secondary task was diminished. Following Haggerty et al. [31] we hypothesized that having participants perform a simple mental task would not diminish the stabilizing effects of head position feedback, but would impair participants ability to perform the mental task [63–65].

Lastly, we hypothesized that experience using the head position feedback system would produce changes in postural control that carried over to trials in which the system was switched off.

## Methods

### Participants

Twelve participants (7 males, 5 females) with an average age of 21.83 years took part in the experiment. All participants were 18 years of age or older. Each participant was pre-screened for medical conditions that impaired their balance, vision, hearing, or vibro-tactile sensitivity. All were students from the University of Connecticut. This research was reviewed and approved by the University of Connecticut Institutional Review Board (IRB). The approved protocol number is H20-0102. All participants read a paper copy of an approved consent form and provided signed and dated (written) consent that was witnessed by the experimenter that conducted the experiment. All participants received partial course credit. Participant recruitment for this study occurred between the 5th of February and the 1st of May in 2022.

### Apparatus and setup

For all standing balance trials participants stood on a force platform with their feet placed inside foot outlines drawn on a sheet of paper adhered to the platform surface. A display screen (0.61 m × 0.34 m) was positioned at a distance of 0.80 m from the front edge of the force platform. The monitor was mounted on a stand that allowed its height to be adjusted and for it to be wheeled around. The height of the monitor was adjusted so that the middle of monitor was at each participant's eye height. On eyes-open standing balance trials participants were asked to look at an '+' marked on a post-it note. This fixation target was positioned at eye height on a wall 2.5 m in front of the participants. On these trials the display screen was moved to one side.

During both the cognitive task familiarization trial and during designated and requested rest breaks participants were seated. On these occasions a chair was moved directly behind the force plate, and participants sat with their feet resting on the force plate. During the cognitive task familiarization trial the display screen was lowered to align with the participant's seated eye-height. No sway data obtained during episodes of sitting was included in the final analysis.

Body sway was recorded using an NDI Certus motion capture system (Northern Digital Inc., Waterloo, ON, Canada). The positions of two active optical markers were tracked. One marker was positioned on the back of a head and was held in place with a headband. The second marker was positioned over the base of the sacrum and was attached to a belt. Marker positions were recorded at 240 Hz.

A force plate (AccuSway, AMTI, Watertown, MA, USA) with surface dimension of 0.5 × 0.5 m, was used to record the location of the center of pressure under the two feet. Although this data was recorded, the analysis of this data is not the focus of this paper.

A slider with a travel range of 6 cm was used as a report device for participants. The slider was custom made using a linear slide potentiometer connected to an analogue-to-digital converter (USB-6225, National Instruments, Austin, TX, USA). Participants held the base of the slider in their left hand and adjusted the slider knob with their right hand.

Two C-2 tactors (Engineering Acoustics Inc., Casselberry Florida, USA) were positioned above the ears and were held in place with Velcro under the headband worn by participants. The tactors were computer controlled via an EAI Universal Controller (Engineering Acoustics Inc., Casselberry Florida, USA). Each tactor was lightweight (17 g) and the diameter of the vibrating element was 7.87 mm (Fig 1C).

The various hardware components involved in this research were all coordinated using a custom C/C++ software application developed by the researchers. This application utilized C/C++ Application Programming Interfaces (APIs) developed by the hardware manufacturers. The application was also used to create a visual interface that displayed hardware status information to the researchers and presented instructions to the participants on the display screen. The visual display was created using the SDL2 intermediate level graphics API (https://www.libsdl.org/). The recording of motion capture data, force plate data, and the analogue signal from the slider were handled in independent threads of the application. Motion capture and analogue signal data were recorded at 240 Hz. Force plate data was recorded at 60 Hz. The main thread of the application handled the obtaining of motion capture and analogue signal data from the other threads, the updating of graphical displays, and the updating of the tactors (i.e., setting the frequency and amplitude of vibration). This thread updated at 60 Hz.

**The real-time head position feedback system.** Our system used two tactors (i.e., vibrating elements) placed above the ears on the left and right side of the head (Fig 1A-B). Given this placement, participants could both hear and feel the vibrations produced by the tactors. This created sources of both haptic (tactile) information and acoustic information. Information about medial-lateral head position was created by having the amplitude of vibration of each tactor vary with changes in medial-lateral head position. Changes in head position were measured via a motion capture system (Fig 1D). The specific relationship that was created between head position and tactor vibration amplitude is described by the tactor activation function shown in Fig 1C. The key characteristic of this function is its "trough" shape. The midpoint of the trough was aligned to a set point ($x^*$) that was obtained by calculating an average medial-lateral head position from three standing balance trials. Given this function, if a participant swayed leftwards away from the set point the amplitude of vibration of the left-side tactor increased with lateral distance. Conversely, if a participant swayed rightwards from the set point, the amplitude of vibration of the right-side tactor increased with lateral distance. In each case, as the participant swayed away from the midpoint tactor vibration amplitude varied from a value of 0.0856 mm at the setpoint, to a maximum value of 0.266 mm at the nadir of the trough. The frequency of tactor vibration was set to a constant value of 250 Hz. While we expect the vibration from the tactors to generate balance relevant information detectible by the haptic and auditory perceptual systems, we do not expect such information to be detected by the vestibular system. The semicircular canals, require rotational displacements of the whole head of .7 to 2.0 degrees per second, depending on which axis of rotation is involved ("roll, pitch, and yaw"). The otolith organs, the vestibular subsystem that detects linear head movements, requires whole-head straight line displacements of .8 to 2.13 cm per second, depending on whether the movement is side-to-side, front-to-back, or up-to-down [66]. Head displacements in any direction that are less than the magnitudes listed above are not adequate to displace the fluids or gels in the vestibular apparatus and so are subthreshold for registration [66].

## Design and procedure

**Standing balance trials.** Multiple standing balance trials were performed across the various phases of the experiment. On all standing balance trials, the participants started by standing on a hard foam pad positioned behind the force plate. On a prompt, they then stepped onto the force plate and positioned their feet inside foot tracings drawn on the surface of the force plate. Participants were directed to reposition their feet if they were not closely aligned to the tracings. On eyes-closed standing balance trials the participants were asked to close their eyes, and report when they were ready for

the trial to begin. On eyes-open standing balance trials participants were asked to look at a fixation point on the far wall and report when they were ready for the trial to begin. At the end of the trial participants were asked to step off of the force platform back onto the foam pad.

**Cognitive dual task.** On some trials participants performed a cognitive task. For this task, participants completed an auditory 3-back error identification task. In this task, participants listened to a series of integer numbers being spoken aloud. The task required participants to count the number of times they heard a number that was not three less than the previous number in the series. The speech for each spoken number was generated using the English (United States) voice generated on https://soundoftext.com/. A new number was spoken every two seconds. For example, in a 30 s duration trial, the sequence of numbers heard by the participant may have been 372, 369, 366, 363, 361, 358, 355, 352, 349, 346, 343, 340, 337, 334, 331. In this example, a series of 14 numbers is spoken and one error is present (i.e., 361 is not three less than 363). Each series was created to either have one or two errors. The starting number, the number of errors (i.e., 1 or 2), and the locations of errors in the series were all randomized.

**Procedure.** A schematic of the various phases of the experimental procedure is presented in Fig 2.

**Initial Setup.** The experiment began with participants reading an informed consent form, and the experimenter explaining the motivation and logistics of the experiment. After consent was obtained and participant demographic information was recorded, participants were given grip socks to wear. The experimenter measured the participant's eye height while both standing and seated. This measurement was used to adjust the display screen to eye-height. It was also used to position a fixation point at eye-height for the eyes-open trials. The participant was asked to stand with feet together with one foot either side of a centerline drawn on a sheet of paper that was adhered to the force plate. Once standing the experimenter drew outlines around the participant's feet. The participant was fitted with an elasticated Velcro head band and a belt. The Velcro-backed tactors were positioned over the ears underneath the headband. One motion tracking marker was secured with Velcro onto the headband at the back of the head. A second motion tracking marker was secured with Velcro onto the belt and was positioned in the middle of the back over the base of the sacrum.

**Determining the set point (x*) of the Tactor Activation Function.** Each participant completed three eyes-closed standing balance trials. Participants were told to stand comfortably. Each trial lasted 30 s. For each trial we calculated the mean medial-lateral head position. These three obtained mean values were in turn averaged, and used to establish a measurement of "neutral" head position in the medial-lateral direction. This value was used as a set point (x*) determining the midpoint of the tactor activation function.

**Tactor threshold sensitivity test.** A threshold sensitivity test was performed to examine to what degree participants would be sensitive to the amplitudes of vibration in the tactor activation function. On each trial participants were presented with a series of vibration pulses that increased in amplitude from 0 mm to 0.18 mm in increments of 0.0017 mm. All pulses had a constant vibration frequency of 250 Hz and a pulse duration of 200 ms. Two trials were performed for both the left and right tactors. In these trials, inter-pulse intervals were either 500 ms or 700 ms. The order of presentation of these trials was randomized across participants. Participants were tasked with reporting when they began to feel/hear the vibration of the tactor. Participants made their report using a slider. The slider had markings for the lower, middle, and upper end it the range. Participants were told to position the slider knob at the lower marker if vibration was "not felt", at the middle mark is a vibration was "possibly felt", and at the upper marker if the vibration was "clearly felt".

**Pre-test.** Participants performed one eyes-open and one eyes-close standing balance trial. Participants were tasked with standing as still as possible [31,33,67]. Each trial lasted 30 s.

**Real-Time Feedback Familiarization.** On this trial, participants were given their first experience using the real-time posture feedback system to aid standing balance. For this trial trough width was set to 50 mm. The trial began with eyes-open and the experimenter encouraging the participant to sway to the left and the right so as to experience the full range of the tactor activation function. As the participant swayed the experimenter explained the basic design of the system. The participants were told to try using the patterns of vibration as an aid to standing as still as possible. Once participants were comfortable using the system with eyes-open they were asked to practice with eyes-closed. This trial lasted 60 s.

**Setup of real-time feedback system**
see text for details

**Tactor threshold sensitivity test**
see text for details

**Pre-test**
2 trials, 30 s per trial

**Standing balance conditions***

EO     EC

**Independent Variables**
Information (EO, EC)

**Familiarization with real-time feedback system**
1 trial, 60 s per trial

**Standing balance conditions**

EO+$F_{ML}$
$w$ = 50 mm

**Cognitive Task Familiarization**
1 trial, 30 s per trial

3-back error identification task (seated)

**Training Phase**
3 blocks, 5 trials in each block, 20 s per trial

**Standing balance conditions** in each block*

EC    EC+$F_{ML}$    EC+$F_{ML}$    EC+$F_{ML}$    EC+$F_{ML}$
         $w$ = 10 mm    $w$ = 20 mm    $w$ = 30 mm    $w$ = 40 mm

**Independent Variables**
Trough width, $w$ (10 mm, 20 mm, 30 mm, 40 mm)

**Self-selected trough width test**
1 trial, 120 s per trial

**Standing balance condition**

EC+$F_{ML}$
$w$ = 1-50 mm

**Testing Phase**
6 trials, 30 s per trial

**Standing balance conditions***

EO    EC    EC+$F_{ML}$    EO+CT    EC+CT    EC+$F_{ML}$+CT
                  $w$ = 20 mm                                $w$ = 20 mm

**Independent Variables**
Information (EO, EC, EC+$F_{ML}$)
Task (Standing balance, Standing balance + CT)

**Post-test**
2 trials, 30 s per trial

**Standing balance conditions***

EO     EC

**Independent Variables**
Information (EO, EC)

**Fig 2. Phases of the experimental procedure.**

**Cognitive Task Familiarization Trial.** On this trial, participants were seated. Participants were given instructions presented on a display screen that described the auditory 3-back error identification task (see above for task details). The experimenter read these instructions out loud and then answered any questions participant's had about the task. Once the participant reported that they understood the task and were ready to begin, the experimenter initiated the trial. The participants listened to the series of numbers and at the end of the trial was prompted to verbally report the number of 3-back counting errors they experienced.

**Training Phase.** In order for participants to practice using our real-time posture feedback system to aid their standing balance the participants completed trials in five different conditions. In all conditions, participants were tasked with trying to stand as still as possible. The first condition was an eyes-closed control condition. Across the four remaining conditions feedback was provided while participants stood with their eyes-closed, and the trough width of the tactor activation function was manipulated to be either 10, 20, 30, or 40 mm. Each condition was completed three times for a total of 15 trials. Each trial lasted 20 s.

**Self-selected Trough Width.** Next, we studied the potential ability of participants to perceive how different parameterization of the head position feedback could aid their ability to stand as still as possible. To achieve this, we gave participants the opportunity to control dynamically a trough width parameter by manually adjusting a hand-held slider while standing with eyes closed. As the slider knob was moved from the lowest to the highest end of its range, the trough width ($w$) of the tactor activation function (Fig 1) dynamically changed from 1 mm to 50 mm. A single trial was performed, with participants given 120 s to 1) explore different trough width settings and 2) find a trough width that best aided their ability to stand as still as possible. Participants began the trial with the slider set to the highest end of its range.

**Testing Phase.** In order to test the post-training effects of real-time postural feedback on standing balance, participants performed standing balance trials either with eyes-closed, eyes-open, or eyes-closed aided by feedback. The trough width in the feedback condition was set to 20 mm. These three conditions were performed as a dual task, in which participants completed the auditory 3-back error identification task at the same time as performing the standing balance task, and as a single task where participants performed the standing balance task without simultaneously performing the auditory 3-back error identification task. The crossing of these conditions produced six trials that were performed in a randomized order. In all conditions, participants were tasked with trying to stand as still as possible.

**Post-Test.** The experiment ended with performances of one eyes-open and one eyes-closed standing balance trial presented in random order. The purpose of these trials was to test whether eyes-open or eyes-closed standing balance performance was altered as a function of participants having experienced multiple trials using head position feedback (i.e., examining carry-over effects). This test was achieved by comparing these post-test trials to the baseline standing balance trials performed at the start of the experiment. In both sets of trials, participants were tasked with trying to stand as still as possible.

## Data analysis

The data analysis presented in this paper focuses upon body sway variability. An analysis of center of pressure data is not included in this first analysis. The recorded positions of motion tracking markers placed on the back of the head and over the base of the sacrum were used as measures of head sway and hip sway respectively. The $x$ and $y$ axis coordinates of the motion tracking markers aligned with the medial-lateral and anterior-posterior axes of body sway. Consequently, $x$ and $y$ coordinates were consequently used as measures of medial-lateral and anterior-posterior sway respectively. To aid our interpretation of the effect of head position feedback, medial-lateral head sway was calculated relative to a set point ($x^*$) that defined the midpoint of the tactor activation function.

Prior to calculating the various measures of sway variability, the time series were downsampled to 24 Hz. This was done to aid the estimation of Ornstein-Uhlenbeck model parameters by removing time scales of variation that were unlikely to be associated with the control of posture. Our initial analysis of sway variability involved calculating the standard deviation of all time-series.

We treat the task of "standing as still as possible" as requiring participants to assemble themselves into a postural control system that possesses fixed-point dynamics. To evaluate the degree to which participants demonstrate such dynamics, we use their head position data to estimate parameters of an idealized Ornstein-Uhlenbeck process, composed of a deterministic *drift* term and stochastic *diffusion* term [41,68]. Each term governs the time-rate of change of head position *x* according to:

$$\dot{x} = \lambda\left(x(t) - x^*\right) + Q\Gamma(t) \tag{1}$$

where $\lambda$ is a constant modulating attractor strength, $x^*$ is the position of the fixed-point, $Q$ is a constant modulating the strength of fluctuating forces, and $\Gamma(t)$ is a Gaussian-distributed random variable with $\mu = 0$ and $\sigma = 1$ representing the fluctuating forces. For $\lambda < 0$, the drift term behaves as an attractor in the dynamics of postural fluctuations while for $\lambda > 0$ it behaves as an unstable repeller.

Our approach to estimating parameters $\lambda$ and $Q$ and evaluating model fit is derived from the published methods of Frank and colleagues [41,68]. To estimate the parameters $\lambda$ and $Q$, we used a discrete approximation of the Ornstein-Uhlenbeck process:

$$X(t + \Delta t) = X(t) + h(X)\Delta t + \sqrt{\Delta t D(X)}\,\Gamma(t) \tag{2}$$

where $h(x)$ and $D(x)$ are surrogate functions for the drift and diffusion terms. The method is initially "agnostic" about the functional form of each term. We then derive a data-based estimate of the functions solely in terms of position *x* and time *t*, then fit those estimates to the functional forms in Eq. 1. $H(x)$ and $D(x)$ are derived by rearranging Equation (2) and taking conditional averages according to:

$$\Delta X = X(t + \Delta t) - X(t) \tag{3}$$

$$h\left(X(t) = K\right) = \frac{1}{\Delta t}\langle\Delta X\rangle\,|_{X(t)=K} \tag{4}$$

$$D\left(X(t) = K\right) = \frac{1}{2\Delta t}\left\langle[\Delta X - h(X)\Delta t]^2\right\rangle|_{X(t)=K} \tag{5}$$

with $\langle\Gamma(t)\rangle = 0$ and $\left\langle\Gamma(t)^2\right\rangle = 2$. The averages are conditioned on *x* values within ranges of the standard deviation around the mean, *K*.

To motivate this, consider that a linear fixed-point attractor of the form in Eq. 1 should demonstrate a restoring velocity that increases linearly with deviations from the fixed point. More specifically, we expect that inter-time step intervals $\Delta X$, should be larger for larger deviations from the fixed point, with their sign varying according to the direction of displacement. Treating the fixed point as coordinate $x = 0$ in a one-dimensional space, a negative $\lambda$ value ensures that deviations below the fixed point result in a positive restoring velocity, and deviations above the fixed point result in a negative restoring velocity (Fig 3B). By binning the data into ranges of the standard deviation, we can estimate the restoring forces from the raw $\Delta X$ values and evaluate their fit to the dynamics predicted from the idealized Ornstein-Uhlenbeck process.

Participant data were converted from raw position to displacement from the determined set point $x^*$. The data were then binned into five ranges *K* defined by the *SD* of the time-series, centered on −2, −1, 0, 1, and 2 standard deviations from the mean, with widths of 1 *SD* (e.g., bin 1 was all data points between −2.5 and −1.5 *SD* from the mean). $H(K)$ and

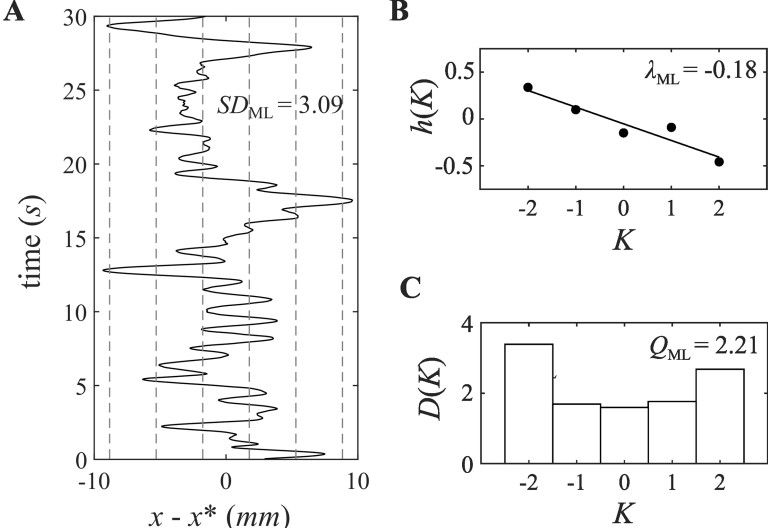

**Fig 3. A) Time series of head position (*x*) measured relative to a predetermined set point (*x\**) for a selected feedback trial (*w* = 20 mm).** Dashed lines demarcate bins used in the calculation of $\lambda_{ML}$ and $Q_{ML}$. Bins are centered around values of $K \cdot s$, where $s$ is the standard deviation of $x$-$x^*$ values, and K is −2, −1, 0, 1, or 2. B) Average fluctuation velocity in each bin ($h(K)$) as a function of bin ($K$). $\lambda_{ML}$ is calculated by taking the slope of the regression fitted to these values. C) Plot of the $D(K)$ as a function of bin ($K$). $D(K)$ is calculated in part by subtracting $h(K)$ from the calculation of fluctuation velocity (i.e., calculating a residual average fluctuation velocity). See text for complete details of the calculation. $Q_{ML}$ is calculated by taking the average of $D(K)$ values.

$D(K)$ estimates were calculated according to Equations 4 and 5. $H(K)$ estimates were regressed onto bins $K$ with the slope of the regression line estimating $\lambda$. $Q$ was expected to be constant with respect to time and displacement, and so it is estimated as the average across the five $D(K)$ bins.

To evaluate the fit of the model to the behavioral data, we compare actual participant variability with the theoretical variability predicted from a drift-diffusion process with the same parameters. Following Bonnet et al., [41] the theoretical $SD$ for an idealized one-dimensional Ornstein-Uhlenbeck process is:

$$SD_{theoretical} = \sqrt{Q/2\lambda} \tag{6}$$

This equation is especially useful for understanding the value of this modelling process, in that it allows us to parse out how two distinct processes—captured via model parameters $\lambda$ and $Q$—shape the simple measure of variability that is commonly used to assess the functioning of postural control.

## Statistical procedures

All statistical analyses were performed using SPSS software version 29.0 (IBM Corporation, Armonk, New York, USA). For each Analysis of Variance (ANOVA) performed, Mauchly's Test of Sphericity was employed to test for violations of the assumption of sphericity. Greenhouse–Geisser corrections were used to adjust the degrees of freedom of the ANOVAs when violations of the assumption of sphericity were discovered. When such adjustments were made, we report the degrees of freedom of that ANOVA to two decimal points. When performed, direct comparisons of the means used a Bonferroni correction to adjust for multiple comparisons. For each presented statistical analysis, we report measures of effect size ($\eta_p^2$) and obtained power ($1 - \beta$).

## Results and discussion

### Tactor sensitivity test

Fig 4 shows how each participant continuously adjusted the report slider in response to the progressively increasing tactor vibration amplitude during each trial of the tactor sensitivity test. All participants can be seen having moved the response slider to report an experience of the tactor vibration being "clearly felt" at a tactor vibration amplitude that was less than the lowest value in the tactor activation function (Fig 1C).

### How did head position feedback affect postural sway?

We began our investigation of how feedback of medial-lateral head position ($F_{ML}$) affected body sway variability by analyzing the data obtained in the test phase. We expected the effects of feedback to be most pronounced in these trials since it was conducted after training. We performed 3 (information) × 2 (task) Repeated Measures ANOVAs on the obtained measures of postural sway variability. The manipulated levels of information independent variable were eyes-closed (EC), eyes-open (EO), and eyes-closed with feedback (EC+$F_{ML}$). The manipulated levels of task independent variable were single task, and dual task. Distributions of medial-lateral head sway values for the three information conditions are shown in Fig 5A.

**Head sway standard deviation (SD).** The standard deviation of head sway in the medial-lateral direction ($SD_{ML}$) was affected by changes in informational support, $F(2, 22) = 14.57$, $p < .001$, $\eta_p^2 = .570$, $1 - \beta = .997$ (Fig 6A). Specifically, pairwise comparisons revealed lower $SD_{ML}$ values in the feedback condition (EC+$F_{ML}$) compared to either the eyes-open (EO) or eyes-closed (EC) conditions, and no difference between the EO and EC conditions. The availability of vibro-tactile head sway information in the EC+$F_{ML}$ condition reduced medial-lateral head sway variability to almost half (53%) that observed in the EC condition. The presence of a cognitive dual task did not affect $SD_{ML}$ values, $F(1, 11) = .302$, $p = .59$, $\eta_p^2 = .027$, $1 - \beta = .079$. or modify observed effect of informational support, $F(2, 22) = .154$, $p = .86$, $\eta_p^2 = .014$, $1 - \beta = .071$. These results confirm our hypotheses that the availability of vibro-tactile information about medial-lateral head position would reduce the variability of head sway in the medial-lateral direction.

In contrast to our results for medial-lateral sway, standard deviation of head sway in the anterior-posterior direction ($SD_{AP}$) was not affected by changes in informational support, $F(2, 22) = 1.30$, $p = .28$, $\eta_p^2 = .106$, $1 - \beta = .251$ (Fig 6B). This

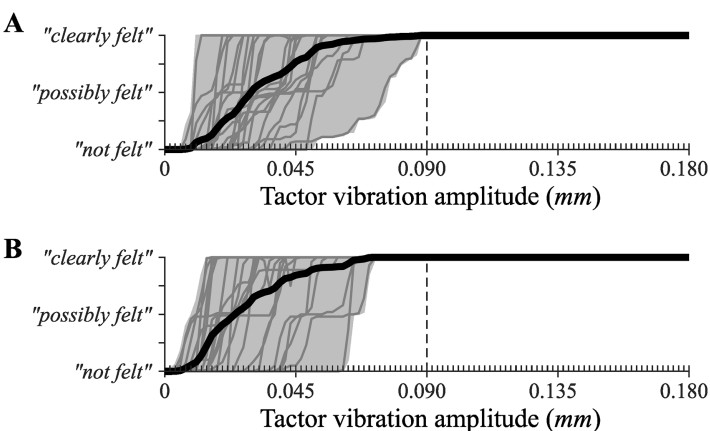

**Fig 4. Responses made during a tactor threshold sensitivity test performed for tactors positioned on the A) left and B) right side of the head.** The thick black line shows the mean response across participants as a function of the experimental manipulation of tactor vibration amplitude. Grey lines show mean response values for individual trials. The grey area outlines the maximum and minimum observed response values. The dashed line highlights the tactor vibration amplitude that was set to be the lowest value in the tactor activation function.

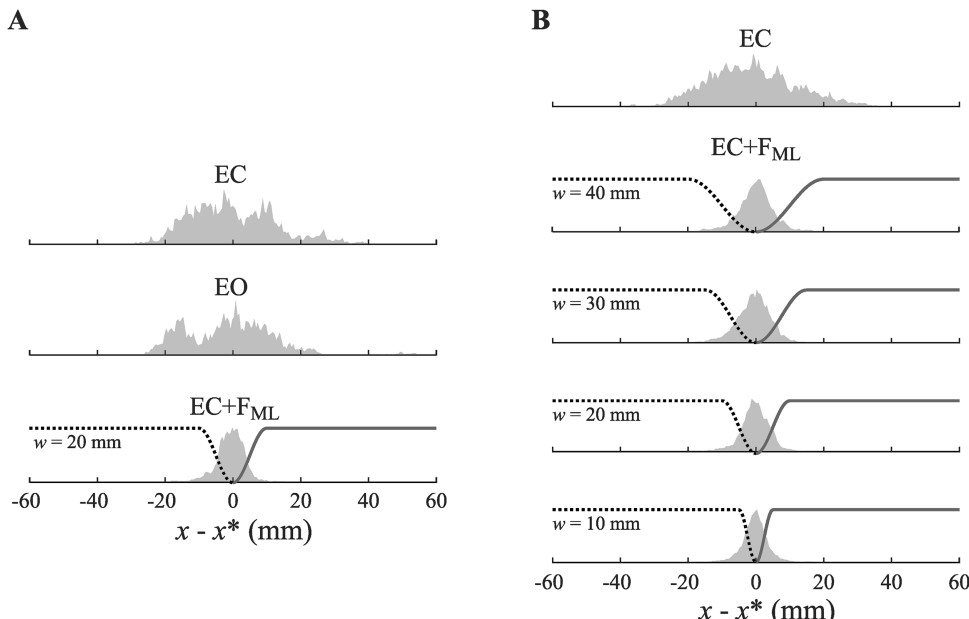

**Fig 5. Distributions of medial-lateral head sway ($x$) for all participants, measured relative the head position set point ($x^*$) shown across A) manipulations of types of information in the test phase, and B) the trough width ($w$) parameter of the tactor activation function in the training phase.**

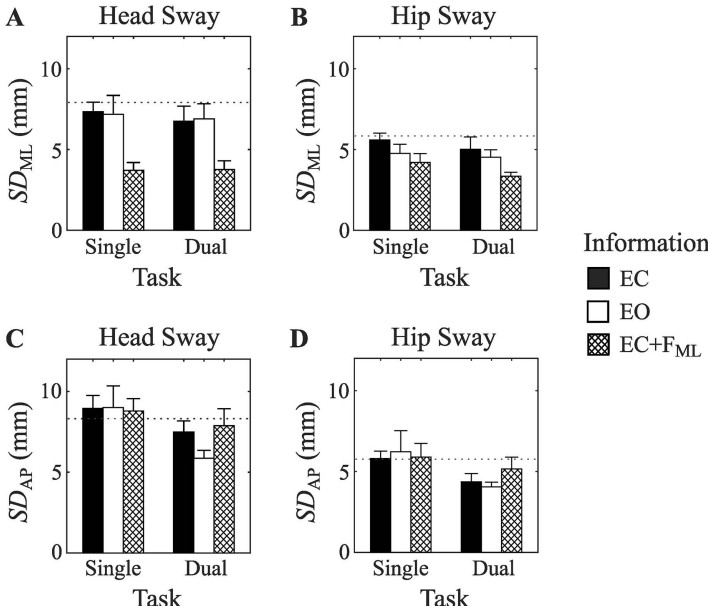

**Fig 6. Test phase movement variability measured as the standard deviation (SD) of head (left panels) and hip (right panels) sway in both the medial-lateral (top panels) and anterior-posterior (bottom panels) sway directions.** Dotted line shows baseline EC condition (performed at the start of the experiment). Error bars show SEM.

is consistent with the fact that in the EC+F$_{ML}$ condition, information was provided about head sway in the medial-lateral direction. Anterior-posterior head sway variability was reduced in the presence of a secondary cognitive task, $F(1, 11) = 5.03$, $p < .05$, $\eta_p^2 = .314$, $1 - \beta = .534$. No interaction between task and informational support was observed, $F(2, 22) = .76$, $p = .48$, $\eta_p^2 = .064$, $1 - \beta = .162$.

**Hip sway standard deviation.** Next, we examined the standard deviation of hip sway in medial-lateral ($SD_{ML}$) and anterior-posterior ($SD_{AP}$) directions. $SD_{ML}$ values were affected by changes in informational support, $F(2, 22) = 5.25$, $p < .05$, $\eta_p^2 = .323$, $1 - \beta = .777$ (Fig 6A). Pairwise comparisons revealed that $SD_{ML}$ values were lower in the EC+F$_{ML}$ condition compared the EC condition. The presence of a cognitive dual task did not affect $SD_{ML}$ values, $F(1, 11) = 3.26$, $p = .10$, $\eta_p^2 = .229$, $1 - \beta = .378$, or modify the observed effect of information, $F(2, 22) = .29$, $p = .75$, $\eta_p^2 = .026$, $1 - \beta = .091$. $SD_{AP}$ values were not affected by changes in informational support, $F(2, 22) = 0.41$, $p = .67$, $\eta_p^2 = .036$, $1 - \beta = .109$ (Fig 6A). $SD_{AP}$ values were reduced in the presence of a cognitive dual task, $F(1, 11) = 7.40$, $p < .05$, $\eta_p^2 = .402$, $1 - \beta = .698$. No interaction between task and informational support was observed, $F(2, 22) = .52$, $p = .60$, $\eta_p^2 = .045$, $1 - \beta = .124$.

These results suggest despite the feedback only providing information about medial-lateral head position, it resulted in body sway variability being reduced at both the head and the hip.

**Modelling sway variability in terms of λ and Q.** To better understand the observed effects of head position feedback on head sway variability, we investigated how observed changes in $SD_{ML}$ might be described as an Ornstein-Uhlenbeck drift-diffusion process. To achieve this, we calculated estimates of two Ornstein-Uhlenbeck model parameters, namely $\lambda_{ML}$ and $Q_{ML}$ (Fig 3).

$\lambda_{ML}$ estimates were affected by changes in informational support, $F(1.17, 12.86) = 18.32$, $p < .001$, $\eta_p^2 = .625$, $1 - \beta = .986$ (Fig 7A). Specifically, pairwise comparisons revealed more-negative $\lambda_{ML}$ values in the feedback condition (EC+F$_{ML}$) compared to either the eyes-open (EO) or eyes-closed (EC) conditions. For this measure, negative values indicate the existence of a location in space that the head tends to drift towards (i.e., a fixed-point attractor), and more-negative values are associated with greater attractor strength. The observed values of $\lambda_{ML}$ were −0.052, −0.024, and −0.169, in the EC, EO, and EC+F$_{ML}$ conditions respectively. These values suggest that in the EC and EO trials, head sway had a slight tendency to drift towards a particular location (i.e., weak attractor strength). In contrast, in the EC+F$_{ML}$ condition, greater attractor strength estimates were observed, with $\lambda_{ML}$ values in the EC+F$_{ML}$ condition being 3.25 times those of the EC condition. The presence of a cognitive dual task did not affect $\lambda_{ML}$ values, $F(1, 11) = .89$, $p = .37$, $\eta_p^2 = .075$, $1 - \beta = .138$, or modify observed effect of informational support, $F(2, 22) = 1.05$, $p = .37$, $\eta_p^2 = .087$, $1 - \beta = .210$.

The second model parameter, $Q_{ML}$, describes the strength of a diffusion process in which fluctuating forces cause the position of the head to randomly walk. Larger $Q_{ML}$ values indicate greater fluctuating forces (e.g., intrinsic noise) in the system and a faster diffusion process. Our estimates of $Q_{ML}$ were affected by changes in informational support, $F(1.29, 14.16) = 4.60$, $p < .05$, $\eta_p^2 = .296$, $1 - \beta = .718$ (Fig 7B). Pairwise comparisons revealed greater estimates of $Q_{ML}$ in the feedback condition (EC+F$_{ML}$) compared the eyes-open (EO) conditions. The observed values of $Q_{ML}$ were 1.689, 1.198, and

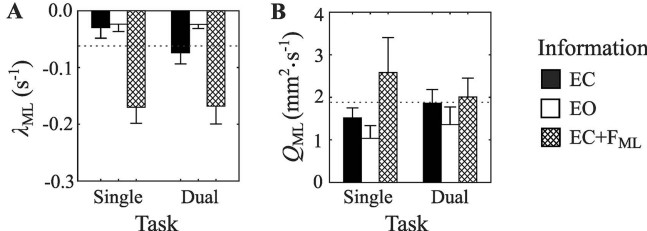

**Fig 7. Test phase medial-lateral head sway variability expressed in estimates of drift-diffusion model parameters.** A) $\lambda_{ML}$ indexes the strength of the drift process, with more negative values associated with greater attractor strength. B) $Q_{ML}$ indexes the strength of the diffusion process, or noise. Dotted line shows values observed in the baseline EC condition performed at the start of the experiment. Error bars show SEM.

2.294, in the EC, EO, and EC+F$_{ML}$ conditions respectively. The presence of a cognitive dual task did not affect $Q_{ML}$ values, $F(1, 11) = .075$, $p = .79$, $\eta_p^2 = .007$, $1 - \beta = .057$, or modify observed effect of informational support, $F(2, 22) = 1.29$, $p = .29$, $\eta_p^2 = .105$, $1 - \beta = .250$.

In sum, the availability of vibro-tactile information about medial-lateral head position established a dynamical process that caused both 1) the position of the head to drift towards the location of the feedback function midpoint, and 2) led to an increase in the random (i.e., not directed towards the attractor) fluctuating forces.

## Validity of modelling head sway variability in terms of λ and Q

Following Bonnet et al. [41], the Ornstein-Uhlenbeck approach predicts a specific relationship between a theoretical standard deviation ($SD_{theoretical}$) and model parameters $\lambda$ and $Q$ (see eq. 1). To test if this prediction holds for our data, we compared the directly measured values of $SD_{ML}$ to predicted values of $SD_{theoretical}$ derived from entering our estimates of $\lambda_{ML}$ and $Q_{ML}$ into eq. 1. For each of the six distinct conditions in the test phase, paired samples t-tests revealed that the difference between $SD_{ML}$ and $SD_{theoretical}$ values did not differ from 0, all $ps > .05$.

## Performance in the cognitive dual task

In the dual task conditions of the test phase participants performed a cognitive task (auditory 3-back error identification task) while also performing the task of standing as still as possible. The cognitive task was performed without any error 50% of the time in the EC+F$_{ML}$ condition, and 83.3% of the time in both the EC and EO conditions. To evaluate cognitive task performance, we calculated a cognitive task error score (CTES) as:

$$\text{CTES} = \left| E_{\text{reported}} - E_{\text{actual}} \right| / \text{n}$$

where $E_{actual}$ is actual numbers of 3-back counting errors in the stimuli, $E_{reported}$ is the number of 3-back counting errors reported by the participant, and $n$ is the total number of 3-back judgements made in the trial. CTES scores were 5.36%, 1.79%, and 1.79% in the EC+F$_{ML}$, EC, and EO conditions respectively. In all three conditions more than half of all obtained CTES scores had a value of 0.0%. This meant that the distributions of CTES scores clearly violated the assumption of normality. We consequently used the non-parametric Wilcoxon Signed-Ranks Test to compare CTES scores across conditions. CTES scores were greater in the EC+F$_{ML}$ condition than in the EC condition, $Z = -2.02$, $p < .05$. No differences in CTES scores were found between the EC+F$_{ML}$ and EO conditions, $Z = -1.89$, $p = .059$, or between the EC and EO conditions, $Z = 0.00$, $p = 1.000$.

## The effect of varying trough-width in the training phase

Given the clear effect of feedback observed in our analysis of the testing phase, we proceeded to an examination of the training phase. We hypothesized that a clear effect of feedback would also be present in this earlier phase. In this phase of the experiment, we manipulated the trough width of the tactor activation function. The marked effect of changes in trough width can be visibly discerned in the distributions of medial-lateral head sway values for the three information conditions is shown in Fig 5B.

Three separate one-way Repeated Measures ANOVAs were used to test whether $SD_{ML}$, $\lambda_{ML}$, and $Q_{ML}$, differed across five conditions. These conditions included one feedback-absent condition (EC) and four feedback-present conditions (EC+F$_{ML}$) in which the trough width of the tactor activation was manipulated to be either 10, 20, 30, or 40 mm.

$SD_{ML}$ values were affected by changes in feedback (Fig 8A), $F(4, 44) = 16.31$, $p < .001$, $\eta_p^2 = .597$, $1 - \beta = 1.000$. Pairwise comparisons revealed $SD_{ML}$ values to be higher in the feedback-absent condition compared to all four of the feedback-present conditions. Pairwise comparisons across the four feedback-present conditions revealed no differences between the trough width conditions. This finding is inconsistent with our hypothesis that reducing trough width would decrease the magnitude of body sway fluctuations.

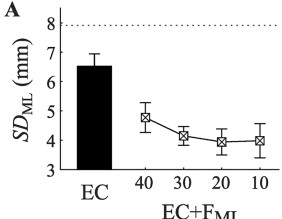 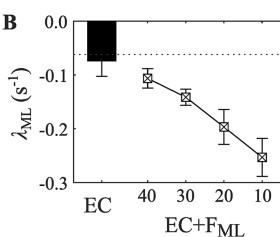 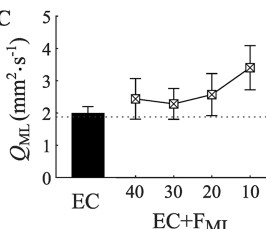

**Fig 8. Training phase medial-lateral head sway variability measured as A) the standard deviation (*SD*) of sway, drift-diffusion model parameter estimates of B) attractor strength ($\lambda_{ML}$), and C) noise ($Q_{ML}$).** Training phase trails include both an eyes closed condition (EC) and four feedback conditions (EC + F$_{ML}$). Feedback trials are shown as a function of the manipulation of vibrotactile feedback trough width to be either 10 mm, 20 mm, 30 mm, or 40 mm. Dotted line shows baseline EC condition (performed at the start of the experiment). Error bars show SEM.

$\lambda_{ML}$ values were affected by changes in feedback (Fig 8B), $F(4, 44) = 11.80$, $p < .001$, $\eta_p^2 = .518$, $1 - \beta = 1.000$. Pairwise comparisons revealed $\lambda_{ML}$ values in two narrowest trough width conditions (i.e., 10 mm and 20 mm) to be more negative (i.e., greater attractor strength) than the feedback-absent (EC) condition. Pairwise comparisons between the four feedback-present conditions revealed more negative $\lambda_{ML}$ values in the narrowest trough width condition (i.e., 10 mm) when compared to both the 30 mm and 40 mm conditions. This finding is consistent with our hypothesis that reducing trough width would increase attractor strength.

$Q_{ML}$ values were affected by changes in feedback (Fig 8C), $F(2.31, 25.40) = 3.54$, $p < .05$, $\eta_p^2 = .244$, $1 - \beta = .645$. Pairwise comparisons revealed that $Q_{ML}$ values in the narrowest trough width conditions (i.e., 10 mm) were greater than the feedback-absent (EC) condition. Pairwise comparisons between the four feedback-present conditions revealed $Q_{ML}$ values to be greater in narrowest trough width condition (i.e., 10 mm) compared to the 40 mm condition.

In sum, in spite of the fact that participants were considered to be learning how to use the device in this phase, we observed clear effects of feedback similar to those observed in the test phase. This suggests that participants were able to use this device to aid their standing still within the 13 trials of initial experience. We also obtained some evidence that the effect of feedback depends upon the trough width of the activation function. Although we did not observe reliable trough-width dependent changes in $SD_{ML}$ values, we did see that a narrowing of trough width is associated with more negative $\lambda_{ML}$ values (i.e., an increase in attractor strength) and an increase in $Q_{ML}$ values (i.e., an increase in noise).

## Do participants prefer a particular trough-width?

In the self-selected trough width trial, participants were tasked with finding the trough width that was best aided their ability to stand as still as possible. We had hypothesized that participants would prefer narrower trough widths. Contrary to this hypothesis, as a group, the participants did not exhibit strong preferences for any particular trough width setting. Of the twelve participants, four selected trough widths between 10 mm and 20 mm, two selected trough widths between 20 mm and 30 mm, three selected trough widths between 30 mm and 40 mm, and three selected trough widths between 40 mm and 50 mm. Of note, none of the participants selected a trough width that was less than 10 mm.

## Does experience of feedback-aided balance affect unaided balance?

We tested the hypothesis that repeated experiences of standing aided by feedback (i.e., the experience gained in both the training and testing phases) would produce a carryover effect affecting standing without feedback at the end of the experiment. To test this hypothesis, we compared the trials performed at the end of the experiment (Post-test Trials) with trials performed at the start of the experiment (Baseline Trials). Both Post-test and Baseline trials included one eyes-closed (EC) and one eyes-open (EO) trial.

We performed 2×2 ANOVAs with Repeated Measures factors of feedback experience (Baseline vs. Post-test) and availability of vision (EC vs. EO) on measures of $SD_{ML}$, $\lambda_{ML}$, and $Q_{ML}$. No effect of feedback experience was observed for any of these measures, $Fs < 1$. Similarly, no interactions between vision and feedback experience were observed, $Fs < 1$.

No effect of vision was observed for $SD_{ML}$, $F(1, 11) = 2.32$, $p = .16$, $\eta_p^2 = .174$, $1 - \beta = .285$. In contrast, effects of vision were observed for both $\lambda_{ML}$ and $Q_{ML}$ estimates. Vision being available was associated with less-negative $\lambda_{ML}$ estimates (EC = −0.068; EO = −0.018), $F(1, 11) = 7.71$, $p = <.05$, $\eta_p^2 = .412$, $1 - \beta = .715$, and lower $Q_{ML}$ estimates (EC = 1.988; EO = 0.945, $F(1, 11) = 17.62$, $p < .001$, $\eta_p^2 = .616$, $1 - \beta = .968$.

## General discussion

We found that young healthy adults who were attempting to stand still with their eyes closed were able to use real-time feedback of head position to decrease their body sway. This effect was enhanced—up to an apparent limit—by narrowing the trough width of the tactor activation function (Fig 8). As predicted, observed effects of head position feedback occurred in the medial-lateral sway direction only (i.e., only in the direction of sway in which feedback was given).

### Effects of head position feedback on the dynamics of standing balance

To further our understanding of the function of head position feedback we analyzed the dynamics of body sway fluctuations using a simple dynamical model designed to capture two basic processes theorized to govern the spatiotemporal trajectory of postural fluctuations. This approach was motivated by a constraints-based and dynamical systems theories of postural control [6,40,69,70]. In these theories, the ability to maintain a desired postural relationship relies on the stabilizing effects of various sources of constraint counteracting the destabilizing effects of various sources of variability [13,36,39]. In our Ornstein-Uhlenbeck model of the dynamics of postural fluctuations we assume that 1) the stabilizing effect of the available sources of constraint are captured through estimates of attractor strength, and that 2) the destabilizing effect of sources of internal perturbations and background/physiological variability in the postural control system [71,72] are captured through estimates of noise.

We hypothesized that the availability of real-time head position feedback would be similar in effect to having a rich source of visual information. Specifically, we predicted that a person aided by head position feedback would show increased attractor strength and/or decreased noise in the dynamics of their head sway fluctuations. Consistent with this prediction we observed increased attractor strength, and that widening/narrowing of trough width in the tactor activation function was associated with decreases/increases in attractor strength respectively. Inconsistent with our prediction, we found that head position feedback increased (rather than decreased) noise. This potentially suggests that the "artificial" informational constraints provided by head position feedback are of a different kind to the "natural" informational constraints provided the visual system. While this is an intriguing claim, it is a premature one. A proper test of this claim would require specifically designed future research to provide a "side-by-side" comparison of these two sources of informational constraint.

While differences in the participant instructions and stimuli make a direct side-by-side comparison between our study and the previous studies examining the effect of changes in richness of visual information [13,14,38,41] problematic, there are points of intersection that may be useful to consider. In these previous studies examining the effect of changes in the availability of visual information for postural orientation, younger (< 25 yo) and older (>65 yo) participants were tasked with "standing comfortably" as though the intention was to do so for an extended period, as when waiting in line or engaged in conversation. In these studies, it was observed that when more visual informational support was made available to older adults instructed to "stand comfortably" an increase in attractor strength coupled with increased noise was identified using a recurrence quantification analysis [13,14,38,41], analogous to the relationship seen in the current study between $Q_{ML}$ estimates and $\lambda_{ML}$ estimates. By contrast, in the current experiment, the instruction was to stand "as still as possible". To the degree, however, that participants interpreted this instruction as requiring the elimination of all possible sway, it is

probable that they attempted to do so by more active co-contraction of muscles to damp out movements around the joints that would otherwise permit movement fluctuations. If a co-contraction control strategy was adopted, then more physiological work was being done and there would necessarily be an elevation of intrinsic noise levels that would increase $Q_{ML}$ estimates. It is well-documented that older adults use co-contraction strategies to enhance postural control more frequently than younger adults engaged in the same tasks [73]. This suggests the possibility that the instructions given in the current experiment led to a significant number of our participants adopting a co-contraction strategy similar to that of older adults in the earlier studies, with similar underlying dynamics. Why then should more informational support, whether visual or vibratory, for postural control result in co-occurrence of a stronger fixed-point attractor and an increase in noise as estimated by our $Q$ metric?

In the current context, it is important to understand that our metric for "noise" is tied to the increase in thermal noise production associated with more work being done in response to more information for the control of postural sway being available. This distinguishes it from "noise" metrics intended to quantize a loss of fidelity during signal transmission. Consider that when more information is available for the perceptual control of actions, more useful work can be done, better ensuring greater congruence with the intended goal, in this case more consistently upright posture. Detectable differences in the workspace of sway control, as were provided by the tactor vibrations, provide the basis for more precisely directed efforts. At the level of the underlying dynamics, this would result in parameterization yielding greater global attractor strength (deeper "well" on the system's manifold) accompanied by an increase in local entropy production ("noise"), literally the heat produced by the physiological work required to satisfy the behavioral goal. The exploitation of information is never free when it is used to move from a higher symmetry state (more random swaying) to a lower symmetry state (more informed and effortful, so less random swaying.) For extensive discussions of the reciprocities between information and action in the context of the attractor dynamics of intentional actions, see [61,74].

In the previous studies examining the effect of changes in the availability of visual information upon the dynamics of postural control [13,14,38,41], it is not clear whether the manipulation of richness of visual information is comparable in "richness" to the information provided by our postural feedback system. In these previous studies, richness of visual information was manipulated by having participants stand in front either a white screen or a depth array. The blank screen created an impoverished visual information condition. The depth array involved three rows of vertically oriented rods and created a condition of comparatively rich visual information. The depth array created a visual environment in which head sway shifted the participant's perspective on depth array and generated optical motion containing balance specific information. Using this apparatus, it would be possible to obtain a method of systematically manipulating the richness of visual information by simply progressively moving the depth array closer to the participant [8]. Based upon our findings we might expect that when we introduce sufficiently rich sources of visual information that increases in attractor strength are accompanied by increases in noise.

## Carry-over effects

In our analyses of both the magnitude of head sway fluctuations and the dynamics of head sway fluctuations we did not observe any evidence of carry-over. Specifically, the experience of using a head position feedback system did not alter how our young healthy participants controlled the posture of their body shortly after the system was switched off. This is consistent with the findings of Balardini et al. [33] who similarly did not observe carry-over effects in measures of the magnitude of postural sway fluctuations in young heathy adults who had received training with a real-time posture feedback system. In contrast to these findings, when individuals with impaired balance have experienced using a real-time postural feedback system, reductions in the magnitude body sway fluctuations have repeatedly been observed after the device was switched off [21]. Balardini et al. [33] did observe carry-over in measures of the frequency profile of postural sway fluctuations. This potentially suggests that carry-over effects are more readily observed in young healthy adults in measures of the *form* of postural control (i.e., the way in which the degrees of freedom of the body are coordinated)

compared to measures of the *function* of postural control (i.e., how effectively the posture of the body is controlled). We plan to examine this possibility in a follow-up paper. Specifically, we plan to analyze the dynamics of center of pressure fluctuations (rather than head sway fluctuations) using the obtained force plate recordings, and the coordination of hip and head. These analyses will allow us to examine whether experience using real-time postural feedback can alter the way in participants control the posture of their body.

### Effects of vision on the dynamics of standing balance

Prior works suggest that the availability of rich sources of visual information together with the capacity of an individual to detect that information should increase attractor strength and/or decreased noise of postural fluctuations [13,41]. The results of our experimental manipulation of vision was not consistent with either of these predictions. We suspect that our experimental setup produced an effective manipulation of the availability of vision (i.e., eyes open vs. eyes closed) but a poor manipulation of the availability of a rich source of visual information. In our study, participants looked at a distal target on the far wall of a largely uncluttered room. This differs markedly to the method used by Kinsella-Shaw and colleagues, of having participants stand in front of a depth grating apparatus (i.e., an array of vertically oriented poles positioned in front of the participant). When standing in front of a such a depth grating apparatus small changes in head position produced clearly discernable changes in optical structure. In future works it would be valuable to perform a side-by-side comparison of the effects of rich sources of "natural" and technology-based information.

### Effects of "dual tasking"

When our participants were tasked with simultaneously performing a simple mental task and using the real-time head position feedback system to stabilize their balance, we observed that 1) the aid-to-balance obtained from real-time head position feedback was not affected, and 2) the ability to perform a simple mental task was diminished (relative to the eyes open and eyes closed control conditions). Both these findings are consistent with the previous findings of Haggerty et al. [31], and taken together suggest that cognitive constraints are a factor in the use of feedback systems for both younger and older adults and should be accounted for when considering the potential application of this technology. It also suggests that participants may tend to prioritize balance tasks over secondary tasks, at least in experimental contexts that have been studied [75].

We further observed that the requirement to simultaneously perform a 3-back error identification task in our study did not affect any of our measures of standing balance in either the eyes open or eyes closed conditions. While this finding is inconsistent with the findings of Pellecchia and collages [48,67] who observed that a 3-back counting task increased the magnitude of postural fluctuations and decreased the attractor strength in the dynamics of postural fluctuations, it does fit with the decidedly mixed results reported in the literature related to this manipulation. While some studies have shown that performing a 3-back counting task increases the magnitude of postural fluctuations [76], others have shown decreases [77,78].

Given that overall dual tasking performance (i.e., the ability to both effectively minimize body sway a perform the cognitive task) was diminished when participants used real-time head position feedback to aid their balance, but not when the used vision to aid their balance, we may be tempted to interpret this as evidence of differences between feedback and vision as informational constraints. This conclusion is premature given the problems with our eyes-open condition that we laid out in the last section.

### Generalizability of findings and modelling approach

The goal of the research presented here was to use a *task dynamic* modelling approach [79,80] to examine how real-time postural feedback can affect a person's ability to control the posture of their body. The task dynamic modelling approach

proposes that a coherent macro-scale pattern of behavior (i.e., a task dynamic) emerges from the interactive effects of task, motor, and informational constraints [4,81]. In our research we purposefully chose a highly constrained task that we believed would lead to the emergence of a simple task dynamic that could be reasonably captured using an Ornstein-Uhlenbeck model. Specifically, in the current study we examine the influence of the information provided by the tactors on the degree of departure from the goal state of zero displacements from the target relationship to the environment. Our experiment was designed with the goal of producing clear and simple test of the hypothesized effects. Our experiment was not designed with the goal of producing results that could be easily generalized to a wide variety of postural control tasks with fewer/different task constraints than those examined here.

Within the ecological-dynamical theory that has been developed around the task dynamic approach, behavioral research findings are predicted to generalize across contexts implicating a similar task dynamic [82–86]. With this principle in mind, we now consider the generalizability of our findings.

**Generalization to the task of standing as still as possible.** In our research we manipulated the informational constraints, and we attempted to control the task and motor constraints. We attempted to control the task constraints through the instructed goal to "stand a still as possible", and we attempted to control the motor constraints by having participants stand with their feet together giving them a narrow medial-lateral base of support. Of especial note, one requirement for satisfying the intention to stand as still as possible is to minimize sway that would displace the head. In our modelling we assume that these controls caused participants 1) to have the goal of organizing their posture so that minimal body movement was generated, and 2) produce a pattern of behavior that approximated the dynamic of a critically damped system as closely as possible, that is, to return the posture of the body to an equilibrium state with as few oscillations around that state as possible. We quantified this task dynamic using an Ornstein-Uhlenbeck model. Given this, we would expect both our findings and modelling approach to be directly generalizable to tasks where there is a clear goal of standing as still as possible. This would include scenarios where greater sway magnitudes are at odds with task demands, e.g., standing with a cup of hot coffee filled to the brim, when trying to blend into the scenery, threading a needle, or taking a picture.

**Generalization to the task of maintaining a comfortable upright stance.** How might we expect the results presented here to generalize to postural tasks differ from the highly constrained task examined here? In our previous work, we have examined a less-constrained task in which participants were asked "stand comfortably" (i.e., in a way that would be comfortable to maintain for an extended period of time) [13,14,37,38,41].

For the task of standing comfortably, our present assumption that 1) effective postural control is associated with minimizing postural fluctuations, and 2) postural fluctuations can be modelled as a simple Ornstein-Uhlenbeck drift-diffusion process, are much less reasonable. That said, in our previous work we have repeatedly found an increase in availability/amount of visual information to be associated with 1) a decrease in the magnitude of postural fluctuations and 2) an increase in the stability of the task dynamics [13,14,38,41]. In one of these studies [41], we further showed that the Ornstein-Uhlenbeck model can be an effective model of the postural fluctuations observed for the task of standing comfortably. In other of our previous studies (Kinsella-Shaw et al., 2006; 2011; 2013), we used a model-free approach in which the task dynamic of standing balance was empirically derived using a phase space reconstruction (PSR) procedure and quantified using variants of recurrence quantification analysis. In these studies, we assumed that a stable task dynamic existed and could be reconstructed using PSR, and we empirically derived measures of the attractor strength ($\lambda$) and noise ($Q$) parameters of that dynamic.

Although some of our previous work suggests that our current analysis/modelling approach can be generalized to the task of standing comfortably, others do not. In our current research, we assumed that the task dynamic reflected the goal of minimizing body sway. Contrary to this assumption, some previous studies implicate a task dynamic for standing comfortably that involves stabilizing center of mass fluctuations rather than body sway fluctuations [87–89]. In our current research, we assumed that the task dynamic had the qualities of a quasistatic fixed-point attractor. Although this

assumption is consistent with some dynamical models of standing balance [36,90–92], it is inconsistent with previous studies that suggest that postural fluctuations are organized around controlling the movement of the center of mass relative to the limits of stability rather than some fixed or drifting set point [93]. Yet other previous studies have suggested that postural fluctuations possess an important exploratory function that serves the purpose of generating information relevant to the perceptual control of action [94,95] and or some grander suprapostural task [96,97]. These studies suggest that in some contexts more sway (rather than less sway) can be functional, and that the relevant task dynamic may be more appropriately modelled in a perceptual-motor phase space rather than purely spatial phase space [49,98].

In sum, the previous literature suggests that there is not a single task dynamic for postural control. This conclusion has two major implications for future work. First, to examine whether real-time feedback can be used to stabilize the task dynamic for less constrained balance tasks it will likely be appropriate to employ the model-free approach mentioned above. Second, if there are marked changes in the task dynamic of less constrained standing balance tasks as a function of subtle changes in the context of constraints, then our simple tactor activation function (that was designed to stabilize a fixed-point dynamics), may be less suited and consequently less effective compared to its implementation here.

## Author contributions

**Conceptualization:** Steven J. Harrison.

**Data curation:** Steven J. Harrison.

**Formal analysis:** Steven J. Harrison, Benjamin De Bari.

**Investigation:** Steven J. Harrison, Ryan Poutre.

**Methodology:** Steven J. Harrison.

**Project administration:** Steven J. Harrison.

**Resources:** Steven J. Harrison.

**Software:** Steven J. Harrison, Benjamin De Bari.

**Supervision:** Steven J. Harrison.

**Validation:** Steven J. Harrison.

**Visualization:** Steven J. Harrison.

**Writing – original draft:** Steven J. Harrison, Jeffrey M. Kinsella-Shaw.

**Writing – review & editing:** Steven J. Harrison , Benjamin De Bari, Jeffrey M. Kinsella-Shaw.

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
