## [Decision Letter · Decision Letter 0]

8 Nov 2024

Dear Dr. harrison,

We look forward to receiving your revised manuscript.

Kind regards,

Dimitris Voudouris

Academic Editor

PLOS ONE

Journal requirements:    When submitting your revision, we need you to address these additional requirements. 1. Please ensure that your manuscript meets PLOS ONE's style requirements, including those for file naming. The PLOS ONE style templates can be found at https://journals.plos.org/plosone/s/file?id=wjVg/PLOSOne_formatting_sample_main_body.pdf and https://journals.plos.org/plosone/s/file?id=ba62/PLOSOne_formatting_sample_title_authors_affiliations.pdf 2. When completing the data availability statement of the submission form, you indicated that you will make your data available on acceptance. We strongly recommend all authors decide on a data sharing plan before acceptance, as the process can be lengthy and hold up publication timelines. Please note that, though access restrictions are acceptable now, your entire data will need to be made freely accessible if your manuscript is accepted for publication. This policy applies to all data except where public deposition would breach compliance with the protocol approved by your research ethics board. If you are unable to adhere to our open data policy, please kindly revise your statement to explain your reasoning and we will seek the editor's input on an exemption. Please be assured that, once you have provided your new statement, the assessment of your exemption will not hold up the peer review process. 3. Please include your full ethics statement in the ‘Methods’ section of your manuscript file. In your statement, please include the full name of the IRB or ethics committee who approved or waived your study, as well as whether or not you obtained informed written or verbal consent. If consent was waived for your study, please include this information in your statement as well.  4. We are unable to open your Supporting Information file [Fig1.eps, Fig2.eps, Fig3.eps, Fig4.eps, Fig5.eps, Fig6.eps and Fig7.eps]. Please kindly revise as necessary and re-upload. 5. Please include captions for your Supporting Information files at the end of your manuscript, and update any in-text citations to match accordingly. Please see our Supporting Information guidelines for more information: http://journals.plos.org/plosone/s/supporting-information. 

Reviewers' comments:

Reviewer's Responses to Questions

**Comments to the Author**

1. Is the manuscript technically sound, and do the data support the conclusions?

Reviewer #1: Partly

Reviewer #2: Yes

2. Has the statistical analysis been performed appropriately and rigorously?

Reviewer #1: Yes

Reviewer #2: Yes

3. Have the authors made all data underlying the findings in their manuscript fully available?

Reviewer #1: Yes

Reviewer #2: No

4. Is the manuscript presented in an intelligible fashion and written in standard English?

Reviewer #1: Yes

Reviewer #2: Yes

Reviewer #1: Data on head displacement was converted into a haptic display in the form of vibrators in a headband. Young adults stood, wearing the sensors and vibrators. When the vibrators were in used, the spatial magnitude of sway was reduced. The authors expend a great deal of effort in analyzing and interpreting this effect. What they do not do is to explain why spatial magnitude is a good (much less ideal) metric for sway. Mechanistic and clinical approaches to postural control traditionally have simply assumed that “more sway” = “less stability”, but that assumption is no longer tenable. The authors also computed some dynamical aspects of sway but they seem to have little independent interest in these measures.

The authors interpret the existing literature on “postural displays” in terms of access to “improved information”, viz., “Real-time postural biofeedback systems are technologies that appear to enhance our ability to detect information about the movements of our body, and as a consequence improve our capacity to control the posture of our body.” An alternative interpretation is that such displays simply provide a different suprapostural task, and that sway is tuned to facilitate performance of such tasks. The phrasing is ambiguous, but in the context of their focus on sway magnitude it may be that the authors assume that people always attempt to minimize sway. In revising, it will be essential to be clear about the authors’ position on this issue, not least because of the very large literature demonstrating that people do not always attempt to minimize sway magnitude.

The focus on “data driven” interpretations of postural control is puzzling given the emphasis found in the work of one of the authors on intentional dynamics which, if I understand that work, includes the claim that behavior happens, in part but always, in relation to goals. The authors might want to take such ideas more seriously in interpreting the present study.

The haptic stimulators were positioned to reflect body sway in the mediolateral axis. Why? Most sway is in the anterior-posterior direction.

In revising, the authors should include some justification for their sample size. Increasingly, such content is mandated in scholarly journals. Ideally, that would include an a priori power calculation, but as it appears to be too late for that the authors might at least include data on a posteriori power.

What is the difference between biofeedback and feedback? Is there any scientific content to the former term? Could we not do as well simply refer to the haptic display as providing feedback?

The stance that the authors assessed was distinctly unnatural. Participants were compelled to keep their feet together, despite the fact that no one does this voluntarily. In other words, the study is about body sway in a non-preferred, unfamiliar stance. In revising, it will be necessary to acknowledge this fact (and its implications for interpretation of the data) and to provide an explicit motivation for it.

Participants stood with their eyes closed. Why was this condition included? What predictions did the authors make about sway with the eyes closed, and what predictions did they make about relations between this condition and sway with the eyes open?

Participants were told to “stand as still as possible”. In revising, it will be necessary to provide an explicit justification for this requirement.

The authors assert that in some conditions participants stood with eyes open and participants “focused only on standing balance”. This statement seems dubious. Presumably, what the authors meant was that the instructions given to participants referred only to stance, as such. But, if I understand the manuscript, the authors collected no data relating to “what participants focused on” and, given that their eyes were open it seems exceedingly likely that they were “focused”, at least in part, on what they could see. Please note also that when the authors refer to “dual task” conditions what they actually seem to be referring to is the instructions given to the participants (e.g., “do this and do that”). Again, there seems to be an implicit assumption that there is only one task (minimizing sway magnitude) unless the experimenter imposes a second task. In revising, please address the large, diverse literature that demonstrates otherwise (e.g., Haddad) and, separately, the use of postural activity to generate information, that is, exploratory sway (e.g., Hajnal, Palatinus). Among other things, the work of those (and other) scholars raises questions about the concept of “dual tasking”. It may be that postural and suprapostural tasks are integrated, such that they are not perceived or controlled as distinct “tasks”.

Attending to the stimulators might simply have been a suprapostural task. That is, participants may have altered their sway so as to better detect the activity of the stimulators, rather than using the stimulators to reduce sway (Riley et al., 1999). If participants had been instructed to “stand comfortably” and told simply to notice (for example) the haptic stimulators, the results likely would have been different. Given this, we can wonder whether the authors’ arguments (and analysis, and conclusions) bear any meaningful relation to 1) ordinary stance (we rarely try to stand as still as possible) or to widespread, natural variations in the suprapostural tasks that are known (reliably, routinely) to influence both the spatial and temporal dynamics of sway. In general, the authors should explain how their study relates to the widespread (and ecologically motivated) argument that sway is not controlled solely—or even primarily—for its own sake but rather, that the avoidance of falling is integrated with the use of bodily movement in support of performance on suprapostural tasks.

The Method section should be revised to provide much greater clarity about the total number of trials, the sequence of trials, the ordering of trials, and the duration of trials.

The final section of the Introduction, “Current Study” does nothing to motivate most of the experimental design. For example, the use of both standing and seated trials, and the use of both eyes open and eyes closed trials.

Riley, M. A., Stoffregen, T. A., Grocki, M. J., & Turvey, M. T. (1999). Postural stabilization for the control of touching. Human Movement Science, 18, 795-817.

Reviewer #2: Additionally, regarding data availability in accordance with PLOS ONE’s policy, I responded "no" to the platform's question on whether data has been made publicly available, as it is phrased in the past tense ("Have the authors made all data underlying the findings in their manuscript fully available?"). While you have not yet deposited the data, I appreciate your commitment to making it publicly available upon acceptance. As the platform does not allow further clarification, I have conveyed this information directly to the editor.

**Do you want your identity to be public for this peer review?** For information about this choice, including consent withdrawal, please see our Privacy Policy

Reviewer #1: No

Reviewer #2: No

---

## [Author Response · Author response to Decision Letter 1]

9 Jun 2025

Response to Editor Journal Requirements

Editor Comments on Journal Requirements 1:

When completing the data availability statement of the submission form, you indicated that you will make your data available on acceptance. We strongly recommend all authors decide on a data sharing plan before acceptance, as the process can be lengthy and hold up publication timelines. Please note that, though access restrictions are acceptable now, your entire data will need to be made freely accessible if your manuscript is accepted for publication. This policy applies to all data except where public deposition would breach compliance with the protocol approved by your research ethics board. If you are unable to adhere to our open data policy, please kindly revise your statement to explain your reasoning and we will seek the editor's input on an exemption. Please be assured that, once you have provided your new statement, the assessment of your exemption will not hold up the peer review process.

Author Response:

We are working on a method of data sharing.

Editor Comments on Journal Requirements #2:

Please include your full ethics statement in the ‘Methods’ section of your manuscript file. In your statement, please include the full name of the IRB or ethics committee who approved or waived your study, as well as whether or not you obtained informed written or verbal consent. If consent was waived for your study, please include this information in your statement as well.

Author Response:

We have modified the text of our ethics statement and in our method section so that the text matches and contains the required information.

Editor Comments on Journal Requirements 3:

We are unable to open your Supporting Information file [Fig1.eps, Fig2.eps, Fig3.eps, Fig4.eps, Fig5.eps, Fig6.eps and Fig7.eps]. Please kindly revise as necessary and re-upload.

Author Response:

We have reexported these files.

Editor Comments on Journal Requirements 4:

Please include captions for your Supporting Information files at the end of your manuscript, and update any in-text citations to match accordingly. Please see our Supporting Information guidelines for more information: http://journals.plos.org/plosone/s/supporting-information.

Author Response:

NA

Responses to Review #1

Reviewer #1: Comment 1

Data on head displacement was converted into a haptic display in the form of vibrators in a headband. Young adults stood, wearing the sensors and vibrators. When the vibrators were in used, the spatial magnitude of sway was reduced. The authors expend a great deal of effort in analyzing and interpreting this effect. What they do not do is to explain why spatial magnitude is a good (much less ideal) metric for sway. Mechanistic and clinical approaches to postural control traditionally have simply assumed that “more sway” = “less stability”, but that assumption is no longer tenable. The authors also computed some dynamical aspects of sway but they seem to have little independent interest in these measures. The authors interpret the existing literature on “postural displays” in terms of access to “improved information”, viz., “Real-time postural biofeedback systems are technologies that appear to enhance our ability to detect information about the movements of our body, and as a consequence improve our capacity to control the posture of our body.” An alternative interpretation is that such displays simply provide a different suprapostural task, and that sway is tuned to facilitate performance of such tasks. The phrasing is ambiguous, but in the context of their focus on sway magnitude it may be that the authors assume that people always attempt to minimize sway. In revising, it will be essential to be clear about the authors’ position on this issue, not least because of the very large literature demonstrating that people do not always attempt to minimize sway magnitude.

Author Response:

Your points here are well taken. We agree with the essence of all of them. We agree with you on the importance of task constraints. We agree with you that the equation “more sway” = “less stability” does not constitute a ubiquitous organizing principle for the human postural control system, and that people do not always attempt to minimize sway. We also agree that there is compelling data in the literature showing that increasing postural sway can be a way of improving information detection in the context of some tasks. That said, we believe that this is not likely in our experimental task where the explicit instructions are to maintain as fixed an orientation to the environment as possible, with the feet in a prescribed location and the head (gaze) fixated at a prescribed target. We believe that one requirement for satisfying this intention is to minimize sway that would displace the head.

Given this context, we can now answer a question you had in your comment, namely, why our chosen measurements of body sway and our chosen measures of the dynamics body sway are interesting and relevant metrics. In our experiment, and in a subset of everyday tasks, the constraints of the task operate such that a person’s goal is to organize one’s posture so that minimal body movement is generated. Expanding this logic to our use of an Ornstein-Uhlenbeck model of the dynamics of postural fluctuations, the constraints of the task operate such that a person’s goal is to approximate the dynamics of a critically damped system as closely as possible, by returning to equilibrium with as few oscillations around it as possible. This would be the optimal postural dynamics for any scenario where greater sway magnitudes were at odds with task demands, e.g., standing with a cup of hot coffee filled to the brim or when trying to blend into the scenery.

We interpret your comment here as a suggestion that these broader theoretical issues be made clear to a reader of our paper. With this end in mind, we have added a new section in the discussion titled: “Generalizability of findings and modelling approach”. The purpose of this section is to summarize the logic of our specific task and the task dynamic modelling approach in a way that provides a (hopefully clear) conceptual framework for readers to consider how to generalize our findings.

Reviewer #1: Comment 2

The focus on “data driven” interpretations of postural control is puzzling given the emphasis found in the work of one of the authors on intentional dynamics which, if I understand that work, includes the claim that behavior happens, in part but always, in relation to goals. The authors might want to take such ideas more seriously in interpreting the present study.

Author Response:

We are not entirely clear on what you are referencing when you express your concern that our manuscript has a “focus on “data driven” interpretations of postural control”. It is not a phrase we used. We interpret your comment to mean that you believe that our paper could be improved with a clearer presentation of the theory that frames the motivation for our research and the interpretation of our research findings. We believe that our new discussion section on “Generalizability of findings and modelling approach” does some useful work in this regard. It appears that you would like us to much more explicitly characterize the task constraints at play in our research. To do this, we have modified the current study section of our paper. We added a statement about what the overarching intention is, that is maintaining a posture that minimizes head movement relative to an eye-height visual target whether vision is available or not. In the current context, satisfying the task demands is synonymous with satisfying the intention, so any actions/behaviors that are consistent with the instructions to participants are reasonably assumed to be in service of what is intended.

The modified text reads as follows:

“In the current study, we examined how real-time feedback of medial-lateral head position (fig 1) would affect both the magnitude and dynamics of body sway fluctuations for young healthy adults instructed to “stand as still as possible”. Participants tried to maintain a posture with minimal head movement either with their eyes closed or with their eyes open gazing at a target presented at a fixed distance at the height of their eyes. Thus, the overarching intention across both these conditions was to maintain a maximally stable head orientation to the environment.”

Reviewer #1: Comment 3

The haptic stimulators were positioned to reflect body sway in the mediolateral axis. Why? Most sway is in the anterior-posterior direction.

Author Response:

We now include a description of the rationale for this design choice in the revised manuscript. It appears in the current study section of the introduction, and reads as follows:

“We provided feedback of medial-lateral head position using tactors positioned above the ears on the left and right side of the head (Fig 1A). Our choice to examine this positioning, rather than an anterior-posterior tactor placement, was motivated in-part by a wish to avoid the complications of introducing potential effects of vibration sensitivity asymmetries into our experiment. Vibration sensitivity asymmetries would likely have existed had we positioned the tactors on the anterior and posterior surfaces of the body (Corniani & Saal, 2020; Weinstein, 1968). The potential to modify postural sway in the medial-lateral direction is of particular interest to us for the following reasons: Changes in medial-lateral sway control has been reported as more predictive of an elevated risk of falls in older adults and neurologically impaired populations (e.g., patients with multiple sclerosis) compared to changes in anterior-posterior sway control (Lord et al., 1999; Maki et al., 1994; Melzer et al., 2004; Sidaway et al. 2022; Stel et al., 2003). The effective regulation of medial-lateral body sway is known to be important for supporting basic human functions such as transitioning between standing and walking (i.e., step initiation in bipedal gait requires controlled lateral weight shifting) (Herman et al., 1973; Mann et al., 1979) and precise aiming (Balasubramaniam & Turvey, 2000). Lastly, even healthy younger adults, studied here, whose risk of injurious falls is considered minimal have been observed to show increased medial-lateral sway variability when visual support is significantly diminished or excluded (Kollegger et al., 1992; Rugelj et al., 2014).”

Reviewer #1: Comment 4

In revising, the authors should include some justification for their sample size. Increasingly, such content is mandated in scholarly journals. Ideally, that would include an a priori power calculation, but as it appears to be too late for that the authors might at least include data on a posteriori power.

Author Response:

We agree with the reviewer that a priori power calculation has a valuable place in research. We also appreciate the point that in hindsight, such power calculations could have been a nice addition to our approach. At your request, we now additionally report posteriori power (i.e., 1 � �) in the reporting of each of our statistical analyses.

Reviewer #1: Comment 5

What is the difference between biofeedback and feedback? Is there any scientific content to the former term? Could we not do as well simply refer to the haptic display as providing feedback?

Author Response:

Our choice to use biofeedback rather than feedback was motivated by its in related papers as a term connoting the use of feedback to control certain bodily processes that normally happen involuntarily. Our use of the term was also selected as a means of differentiating the feedback studied here from the very many uses of the term feedback in the literature (i.e., performance feedback at the end of a trial). We have no problem designating what the tactors provide as “feedback”. One benefit of this change is that in clinical parlance the usage of the term biofeedback is reserved for when information is being provided about an underlying and often autonomic process in the hopes that doing so would allow the patient to learn to control it more effectively. To make this change we have added clarifying language (e.g., real-time feedback of medial-lateral head position) in order to minimize possible confusion of the reader regarding the meaning of the word feedback.

Reviewer #1: Comment 6a

The stance that the authors assessed was distinctly unnatural. Participants were compelled to keep their feet together, despite the fact that no one does this voluntarily. In other words, the study is about body sway in a non-preferred, unfamiliar stance. In revising, it will be necessary to acknowledge this fact (and its implications for interpretation of the data) and to provide an explicit motivation for it.

Author Response:

In our new discussion section on “Generalizability of findings and modelling approach” we discuss the motivation for our experimenter introduced constraints. As we discuss in this section, what we take to be relevant to the generalizability of our results is not how someone is standing, but the emergent task dynamics promoted by the context of constraints (including how they are standing).

To directly address your request to present the logic of our choice to constrain stance, we have included the following text in the current study section of the introduction:

“Additionally, participants were to maintain the maximally achievable stable head orientation while standing with a narrow base of support, feet together symmetrically relative to the mid-line of the force platform. This postural constraint was introduced to minimize the range over which sway could occur before destabilizing a participant’s posture, moving them away from the intended head orientation. Standing in a symmetrical, “feet together” stance is a common component of clinical balance testing. A narrowed stance, reducing the base of support’s dimensions in the frontal plane has been shown to elevate control demands of stabilizing posture under a variety of conditions, including in the absence of visual support. Given that the participants were all healthy, young adults with no histories of balance impairments, this level of challenge was imposed to increase the likelihood that providing additional information for control, i.e., real-time vibrotactile feedback, would yield measurable differences in sway dynamics and magnitudes.”

In the context of natural settings, many circumstances arise in everyday settings when environmental constraints on actors compel them to adopt a wide range of foot placements, including standing on one foot, in order to satisfy the intention of maintaining upright posture with a given orientation to the environment. People are often required to assemble a narrower than normally preferable base of support while there is the concurrent goal of establishing a particular line of sight in the environment. Traversing a series of supporting surfaces that provide potential points of observation, such as stepping stones, ladders, stairs, etc.,all can require foot placement that differs from what a person would adopt if the goal were to simply stand on a planar surface of support with dimensions greater than the distances between their lower limbs. Additionally, people will often choose to stand with their feet very close together when in a crowd or when the goal is to draw themselves up to their full height for any reason, including improving lines of sight. Any stance that exploits what the local layout of surfaces afford such that an intention is satisfied constitutes a “natural” postural strategy.

Reviewer #1: Comment 6a

Participants stood with their eyes closed. Why was this condition included? What predictions did the authors make about sway with the eyes closed, and what predictions did they make about relations between this condition and sway with the eyes open?

Author Response:

Although we discussed our expectations regarding this manipulation in the discussion section of our original manuscript, we omitted explicitly stating our predictions in the current study section. We have corrected this error. In the current study section we now cover the log

---

## [Decision Letter · Decision Letter 1]

22 Jul 2025

Dear Dr. harrison,

Thank you for submitting your manuscript to PLOS ONE. I have now received the report of one of the original reviewers. The other reviewer is unavailable to submit their comments any time soon. In consideration of your work and for a timely decision, I went through the manuscript once more, and further assessed whether the revised version addressess this reviewer's concerns. Please find below some further comments that should be addressed in a new version. I will then submit a decision to the editorial office. 

We look forward to receiving your revised manuscript.

Kind regards,

Dimitris Voudouris

Academic Editor

PLOS ONE

Journal Requirements:

**Additional Editor Comments:**

1. In addressing the comments by the original reviewers, some parts became quite long. For instance, the discussion of the dual-tasking in the Introduction (lines 273-293) can be shortened so that your manuscript can maintain its focus. Likewise, the Discussion has expanded considerably (e.g., lines 987-1108), and I would recommend to make these new parts more concise. In addition, the discussion of your main finding that head position feedback increases noise is important but some speculations become rather long (e.g., thermal noise). Please shorten parts that are rather speculative or can become more concise in order to maintain a sharp focus on your main findings. 

2. Related to these, please review your reference list and assess whether it is necessary to cite 128 studies to back-up the claims and connect with the previous work. This list feels quite long at the moment. In addition, some references may need more details to comply with the standards of citing a book or a book chapter (e.g., #113 and #115). Reference #74 has also a typo (year is mentioned twice). 

3. In lines 158-160 it is mentioned that larger λ represent greater attractor strength. However, in line 674 it is mentioned that "more-negative values" are associated with greater attractor strength. These are inconsistent as "more negative values" are "smaller", not "larger". Please clarify. 

4. Line 162 mentions that larger Q values indicate a faster diffusion process. Is this true? As far as it seems to me, Q indicates diffusion but this diffusion can occur in many ways (e.g., fast small fluctuations or many larger/slower ones). Please clarify whether Q indeed captures directly the speed of the fluctuation. 

5. It would be helpful if you would add a few comments as to whether the vibrations above the ears could impact the vestibular system, which in turn can influence balance.

6. Please report whether there were any differences between EO and EC in the ANOVA presented in lines 623-627. This comparison is misisng, suggesting that there is no effect. However, as there is a specific hypothesis about the role of vision on body sway (lines 234-237), this test should become clear. In addition, I would recommend to keep statistical procedures for testing the hypotheses as these are outlined in the introduction. Related to this, it is unclear why the effects of informational support on __*SDap* __ should be examined with an ANOVA (lines 634-641) considering that there is no hypothesis about body sway in AP direction. If there is a hypothesis, please add this in the Introduction and justify it. Otherwise, it would be suffiecient to show the related data in a figure (which seems to be missing). 

7. There are some tests in lines 727-729 that appear to be non-parametric. Please justify the use of non-parametric tests in certain cases. 

8. Finally, related to comment 6, I suggest keeping the part in lines 730-779 shorter and more concise. One possibility would be to present the three main ANOVAs in a unified manner and convey a single message. For instance, that "the feedback condition generally reduced the SD, and led participants to drift to their fixed point with greater noise". This is just a quick idea and far from optimal, but my point is to present these exploratory analyses in a shorter and more accessible form. 

9. Figure 5, 6, 7 seem to br missing from the manuscript. Please provide them in the next submission. 

Minor

Line 448, a preposition seems to be missing: "..to sway __to__ the left..."?

Line 524: "Guassian" should be "Gaussian"?

Line 612: it would help if you would remin the reader about the levels of "information" and of "task". 

Line 801: this effect is mentioned in the previous line, I am not sure if it is necessary to be repeated. 

Lines 859-861: this statement requires a citation. Please consider whehter you can use any already existing citations to back this up, in light of an earlier comment related to the number of cited studies. 

Lines 1005 and 1022, these sentences read odd. I guess these are supposed to be sub-headers, so in this case please format the accordingly. Otherwise, please integrate them more smoothly in the text. 

Reviewers' comments:

Reviewer's Responses to Questions

**Comments to the Author**

Reviewer #1: All comments have been addressed

2. Is the manuscript technically sound, and do the data support the conclusions?

Reviewer #1: Yes

3. Has the statistical analysis been performed appropriately and rigorously?

Reviewer #1: Yes

4. Have the authors made all data underlying the findings in their manuscript fully available?

Reviewer #1: Yes

5. Is the manuscript presented in an intelligible fashion and written in standard English?

Reviewer #1: Yes

Reviewer #1: The authors have done a good job in revising. I continue to type to meet the minimum character count.

**Do you want your identity to be public for this peer review?** For information about this choice, including consent withdrawal, please see our Privacy Policy

Reviewer #1: No

---

## [Author Response · Author response to Decision Letter 2]

3 Sep 2025

A formatted version of the response to reviews below has been uploaded with this submission

Introduction to our response to reviews

Dear Editor

We are immensely grateful for your considerable efforts in carefully reading and reviewing our paper. Your comments and suggestions were very useful and have motivated multiple changes that have significantly improved the manuscript yet again.

We describe the work we have done in response to your suggestions in detail below.

Thank you for your efforts

With thanks

Steven Harrison

Response to Editor Comments

Editor Comment 1:

In addressing the comments by the original reviewers, some parts became quite long. For instance, the discussion of the dual-tasking in the Introduction (lines 273-293) can be shortened so that your manuscript can maintain its focus. Likewise, the Discussion has expanded considerably (e.g., lines 987-1108), and I would recommend to make these new parts more concise. In addition, the discussion of your main finding that head position feedback increases noise is important but some speculations become rather long (e.g., thermal noise). Please shorten parts that are rather speculative or can become more concise in order to maintain a sharp focus on your main findings.

Author Response:

We have dramatically shortened the section on dual-tasking in the introduction. To reduce the length of the discussion in the area you highlighted (i.e., lines 987-1108 in the track changes version of the first revision) we have completely removed the section on “What are we studying here?”. We believe that this section is not necessary for the paper. We have looked at the section on “Generalizability of findings and modelling approach”. We did not find any opportunities for shortening this section that did not negatively affect our ability to clearly lay out the logic of how we believe our findings could be generalized. Regarding the paragraph on thermal noise, we again did not find any opportunities for shortening this section that did not make the logic we were presenting opaque. We are hesitant to remove this section since noise is one of our main dependent variables and the logic in this paragraph helps frame a clear interpretation of our findings.

Editor Comment 2:

Related to these, please review your reference list and assess whether it is necessary to cite 128 studies to back-up the claims and connect with the previous work. This list feels quite long at the moment. In addition, some references may need more details to comply with the standards of citing a book or a book chapter (e.g., #113 and #115). Reference #74 has also a typo (year is mentioned twice).

Author Response:

We have corrected the Weinstein reference (Previously #74).

We have corrected the Riccio reference (Previously #113).

We have corrected the Hajnal reference (Previously #115).

We have also made minor corrections to multiple other references

Lastly, we have carefully gone through the paper and examined our references. We have identified and removed all references that appear redundant. We have removed a total of 17 references.

Editor Comment 3:

In lines 158-160 it is mentioned that larger λ represent greater attractor strength. However, in line 674 it is mentioned that "more-negative values" are associated with greater attractor strength. These are inconsistent as "more negative values" are "smaller", not "larger". Please clarify.

Author Response:

We have corrected this confusing wording. It is the case that more-negative values are associated with greater attractor strength.

Editor Comment 4:

Line 162 mentions that larger Q values indicate a faster diffusion process. Is this true? As far as it seems to me, Q indicates diffusion but this diffusion can occur in many ways (e.g., fast small fluctuations or many larger/slower ones). Please clarify whether Q indeed captures directly the speed of the fluctuation.

Author Response:

In the previous work (Bonnet et al) that developed and applied this modelling approach, a larger Q value is interpreted as indicating a more rapidly diffusion process. We have added this citation to the end of this sentence so that the reader understands where this interpretation of Q comes from.

Editor Comment 5:

It would be helpful if you would add a few comments as to whether the vibrations above the ears could impact the vestibular system, which in turn can influence balance.

Author Response:

We do not expect the tactor vibrations to influence balance. Our logic for this conclusion is as follows: Stated simply, the action of the tactors does not impose forces on the head that result in either rotary or linear displacements, so do not provide stimulation for the vestibular apparatus. The semicircular canals, require rotational displacements of the whole head of .7 to 2.0 degrees per second, depending on which axis of rotation is involved (“roll, pitch, and yaw”). The otolith organs, the vestibular subsystem that detects linear head movements, requires whole-head straight line displacements of .8 to 2.13 cm per second, depending on whether the movement is side-to-side, front-to-back, or up-to-down. Head displacements in any direction that are less than the magnitudes listed above are not adequate to displace the fluids or gels in the vestibular apparatus and so are subthreshold for registration.

We have added this information and a relevant citation to the paper.

Editor Comment 6:

Please report whether there were any differences between EO and EC in the ANOVA presented in lines 623-627. This comparison is misisng, suggesting that there is no effect. However, as there is a specific hypothesis about the role of vision on body sway (lines 234-237), this test should become clear. In addition, I would recommend to keep statistical procedures for testing the hypotheses as these are outlined in the introduction. Related to this, it is unclear why the effects of informational support on __SDap__ should be examined with an ANOVA (lines 634-641) considering that there is no hypothesis about body sway in AP direction. If there is a hypothesis, please add this in the Introduction and justify it. Otherwise, it would be suffiecient to show the related data in a figure (which seems to be missing).

Author Response:

We have added information about the pairwise comparison between EO and EC in the ANOVA of SDML values presented in lines 623-627 (your line numbers referring to the track changes version of the first revision). In regards to the effect of SDap, we had stated the hypothesis that “We expected to observe these predicted effects in the medial-lateral sway direction only (i.e., only in the direction of sway in which feedback was given)”. We have edited the language here to more explicitly and effectively highlight our hypothesis of directional specificity.

Editor Comment 7:

There are some tests in lines 727-729 that appear to be non-parametric. Please justify the use of non-parametric tests in certain cases.

Author Response:

In our previous manuscript we justified our use of a non-parametric test with the following statement: ”Given the high percentage of trials in which no errors were made, we used Wilcoxon Signed-Ranks Tests to compare CTES scores across conditions.” We have expanded upon this statement to make the logic more explicit. We now state that: “In all three conditions more than half of all obtained CTES scores had a value of 0.0 %. This meant that the distributions of CTES scores clearly violated the assumption of normality. We consequently used the non-parametric Wilcoxon Signed-Ranks Test to compare CTES scores across conditions.”

Editor Comment 8:

Finally, related to comment 6, I suggest keeping the part in lines 730-779 shorter and more concise. One possibility would be to present the three main ANOVAs in a unified manner and convey a single message. For instance, that "the feedback condition generally reduced the SD, and led participants to drift to their fixed point with greater noise". This is just a quick idea and far from optimal, but my point is to present these exploratory analyses in a shorter and more accessible form.

Author Response:

To clarify, the analyses in this section are not exploratory analyses. In our introduction we state the hypothesis that “We predicted that reducing trough width would both decrease the magnitude of body sway fluctuations and increase the attractor strength in the dynamics of postural fluctuations.” Given this hypothesis, it is necessary for us to report both the main effect and post-hoc pairwise comparisons. We have added some clarifying sentences to remind the reader of these hypotheses.

Editor Comment 9:

Figure 5, 6, 7 seem to be missing from the manuscript. Please provide them in the next submission.

Author Response:

We are not sure how they were ommitted. We will pay special attention when uploading them this time.

Minor Comment from Editor 1:

Line 431, a preposition seems to be missing: "..to sway __to__ the left..."?

Author Response:

We have added in the missing preposition.

Minor Comment from Editor 2:

Line 524: "Guassian" should be "Gaussian"?

Author Response:

This typo has been corrected.

Minor Comment from Editor 3:

Line 612: it would help if you would remind the reader about the levels of "information" and of "task".

Author Response:

We have inserted a reminder about the levels of task and information.

Minor Comment from Editor 4:

Line 801: this effect is mentioned in the previous line, I am not sure if it is necessary to be repeated.

Author Response:

We believe that these refer to different effects. The paragraph you reference on Line 801 summarizes the main effect of vision. The previous lines summarize the vision x biofeedback interaction effects.

Minor Comment from Editor 5:

Lines 859-861: this statement requires a citation. Please consider whether you can use any already existing citations to back this up, in light of an earlier comment related to the number of cited studies.

Author Response:

We do not have an existing reference for this point. We have inserted one key reference to support it:

Cenciarini, M., Loughlin, P. J., Sparto, P. J., & Redfern, M. S. (2009). Stiffness and damping in postural control increase with age. IEEE Transactions on biomedical engineering, 57(2), 267-275.

Minor Comment from Editor 6:

Lines 1005 and 1022, these sentences read odd. I guess these are supposed to be sub-headers, so in this case please format the accordingly. Otherwise, please integrate them more smoothly in the text.

Author Response:

We have more clearly formatted this as a sub-heading. Our sub-heading formatting in our last submission was hidden by the track changes formatting.

---

## [Editor Report · Decision Letter 2]

23 Sep 2025

Understanding the effects of real-time head position feedback on postural sway in terms of changes in underlying deterministic and stochastic dynamical processes.

PONE-D-24-33781R2

Dear Dr. harrison,

We’re pleased to inform you that your manuscript has been judged scientifically suitable for publication and will be formally accepted for publication once it meets all outstanding technical requirements.

Kind regards,

Dimitris Voudouris

Academic Editor

PLOS ONE
---

## [Editor Report · Acceptance letter]

PONE-D-24-33781R2

PLOS ONE

Dear Dr. Harrison ,

I'm pleased to inform you that your manuscript has been deemed suitable for publication in PLOS ONE. Congratulations! Your manuscript is now being handed over to our production team.

Kind regards,

on behalf of

Dr. Dimitris Voudouris

Academic Editor

PLOS ONE